# Microfacies and Depositional Conditions of Jurassic to Eocene Carbonates: Implication on Ionian Basin Evolution

**Nicolina Bourli** [1,*], **George Iliopoulos** [2], **Penelope Papadopoulou** [2] **and Avraam Zelilidis** [1]

1   Laboratory of Sedimentology, Department of Geology, University of Patras, 26504 Patras, Greece; a.zelilidis@upatras.gr
2   Laboratory of Paleontology, Department of Geology, University of Patras, 26504 Patras, Greece; iliopoulosg@upatras.gr (G.I.); penelpapadop@upatras.gr (P.P.)
*   Correspondence: n_bourli@upnet.gr

**Abstract:** In order to decipher the paleo-depositional environments, during the Late Jurassic to Early Eocene syn-rift stage, at the margins of the Ionian basin, two different areas with exposed long sequences have been selected, Kastos Island (external margin) and Araxos peninsula (internal margin), and were examined by means of microfacies analysis and biostratigraphy. On Kastos Island, based on lithological and sedimentological features, the following depositional environments have been recognized: an open marine/restricted environment prevailed during the Early Jurassic ("Pantokrator" limestones), changing upwards into deep-sea and slope environments during the Late Jurassic and Early Cretaceous (Vigla limestones). The Upper Cretaceous (Senonian limestones) is characterized by a slope environment, whereas during the Paleogene, deep-sea and toe of slope conditions prevailed. In Araxos peninsula, Lower Cretaceous deposits ("Vigla" limestones) were accumulated in a deep-sea environment; Upper Cretaceous ones (Senonian limestones) were deposited in slope or toe of slope conditions. Paleocene limestones correspond to a deep-sea environment. In Araxos peninsula, changes occurred during the Cretaceous, whereas on Kastos Island, they occurred during the Paleocene/Eocene, related to different stages of tectonic activity in the Ionian basin from east to west.

**Keywords:** standard facies zones; microfacies; Ionian basin; Kastos Island; Araxos peninsula; basin evolution

## 1. Introduction

Microfacies analysis of carbonate rocks requires knowledge of modern carbonates and a good understanding of biological and geological changes during Earth history. Many criteria that are important in facies interpretation cannot be recognized sufficiently in small-scale microfacies.

Paleoenvironmental interpretations deriving from microfacies analyses should be controlled by lithological criteria and sedimentary structures evaluated by the high information potential provided by fossils and biogenic structures as well. Microfacies sampling requires an understanding of the meaning of bedding and the respective depositional characters as they are reflected by sedimentary structures [1–4].

Several classification protocols have been proposed for the classification of limestones, taking in mind morphological and/or genetic factors, which emphasize on specific properties, such as color, size of crystals, composition, fabric, and texture. In general, the classification is based on the grain size and fabric properties of the rocks [5]. The most common classifications are: (A) the classification of Folk [6] is based on the classification and fabric maturation of limestones. It is purely morphometric and has limited genetic extensions, regarding the possible level of energy, transport, and deposition of grains. Folk [6] found that most carbonate rocks are composed of: 1. non-skeletal grains, 2. skeletal grains, 3. microcrystalline calcite or micrite, 4. crystalline calcite or sparite as a cement chemically

formed filling the voids between the grains of carbonate rocks. (B) The classification of Dunham [7] is a fabric-based classification, distinguishing between those carbonate rocks whose grains are in contact with or simply coexist within a carbonate filler.

In both classifications, the purpose is not only to classify the rock, but also to provide information on the energy levels of the depositional environment. According to Folk [6], a micrite or biomicrite indicates a low-energy depositional environment while a biosparite indicates high-energy conditions. According to Dunham [7], mudstone and wackestone correspond to a low energy environment, in contrast with packstone and grainstone that refer to high-energy conditions.

Lithology, texture, grain types, fossils and their percentages, sediment structures, color, and fabric character are taken into account for the determination and interpretation of carbonate facies. A detailed description of carbonate facies requires field, as well as microscopic observations to determine it [5]. Therefore, herein the model of Wilson [8] for the determination of the carbonate facies is followed with twenty-four (24) standard microfacies (SMF), corresponding to nine (9) standard facies zones (FZ), from open deep-sea basin environments, the slope, the edge of the platform, and the inner platform [5,9].

The aim of this paper is to define in detail the depositional environments of the carbonate Mesozoic sequence of the Ionian Basin, in the external sub-basin (Kastos Island) and the internal sub-basin (Araxos peninsula), respectively. The target is the vertical and lateral depositional condition changes both internal to the same age and different age formations, and so on the reconstruction of geodynamic conditions during sedimentation and their support to rebuild of the geological map. The achievement of the above was made using microfacies analysis. The taphonomy and paleoecology of macrofauna have also been used in order to reconstruct the depositional environments and the dynamic conditions in different positions across the basin, and to support the previous lithofacies results. Microfossils have been used to provide paleoecological and biostratigraphic data, and the chronostratigraphy of the studied lithological formations particularly from Kastos Island has been reconsidered. The results were used, both for the revision of existing geological maps as well as for the correlation of the different depositional conditions between the studied parts of the basin.

## 2. Geological Setting

### 2.1. Ionian Basin

The Mesozoic-Paleogene Ionian basin is part of the external Hellenides orogen, bounded westwards by the Ionian Thrust and eastwards by the Gavrovo Thrust (Figure 1). The Pre-Apulian (or Paxoi zone) to the west is regarded as the eastern margin of the Apulian platform, in Albania, Croatia, and Italy, where similar rocks occur [10–12].

The sedimentary fill of the basin is sub-divided into pre-rift, syn-rift, and post-rift tectono-sedimentary sequences [12–14] the syn-rift and post-rift sequences are merged into one (syn-rift: [15–17]) (Figure 2):

(i) The pre-rift sequence begins with evaporites (Lower to Middle Triassic) at the base (over 2000 m thick). These deposits evolve upwards into the Middle-Upper Triassic (Ladinian to Rhaetian) "Foustapidima" limestones (50–150 m thick), and the Lower Jurassic (Hettangian to Sinemurian) "Pantokrator" limestones (over 1000 m thick).

(ii) The syn-rift sequence was deposited during the extension and deepening of the Ionian basin, accompanied by its internal structural differentiation into small sub-basins with half-graben geometry, displaying abrupt thickness changes within each other [13,14]. The syn-rift sequence is composed of Upper Jurassic to Lower Eocene deposits. At the base, it consists of the Lower Jurassic (Pliensbachian) pelagic "Siniais" limestones and their lateral equivalent the hemipelagic "Louros" limestones (20–150 m thick). These deposits underlay the Lower to Upper Jurassic (Toarcian to Tithonian) "Ammonitico Rosso," "Limestones with filaments," and "Posidonia beds" (20–200 m thick). Variations in thickness and formation changes across the basin are observed very often due to the half-graben geometry, thus different basin depths are commonly observed.

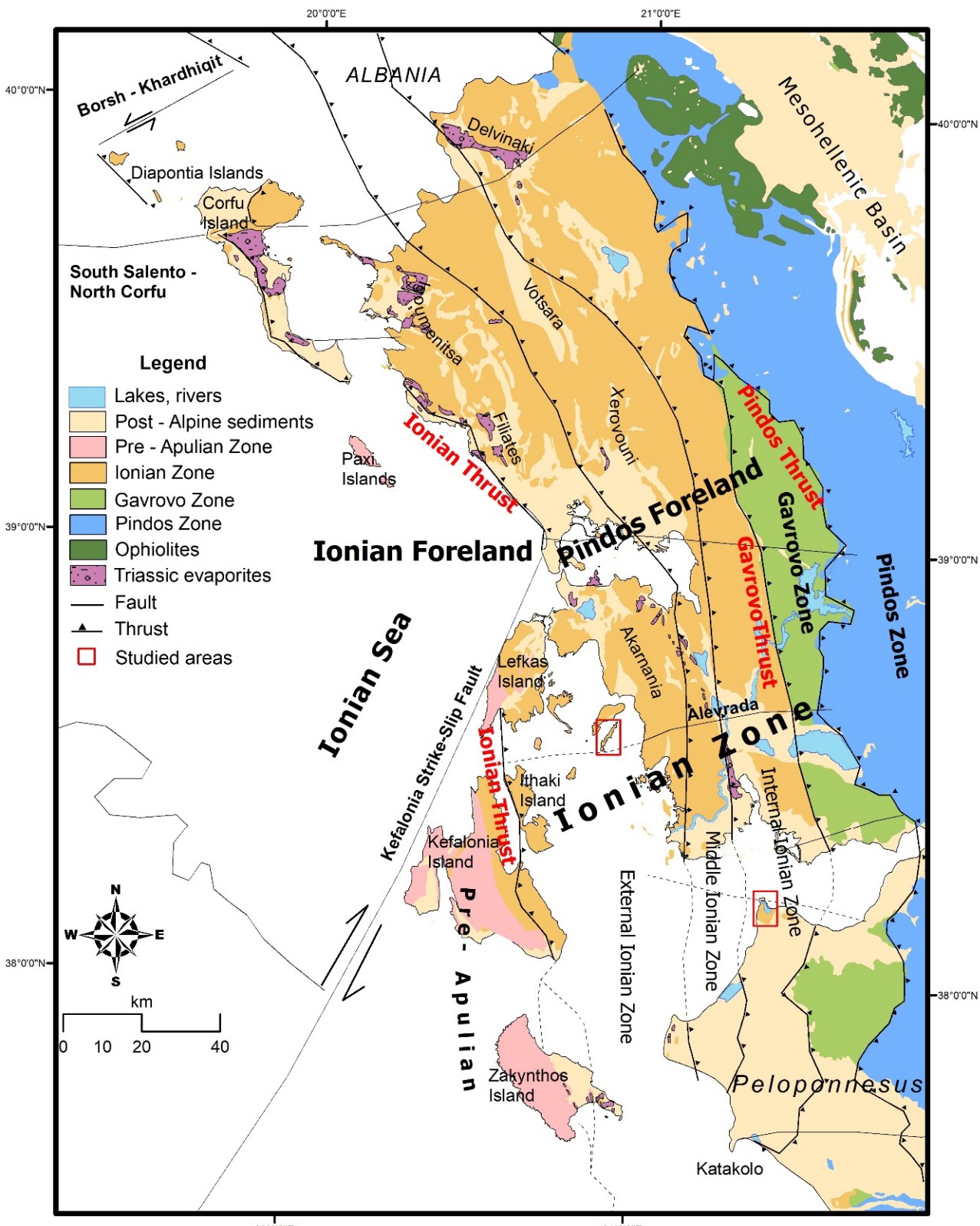

**Figure 1.** Geological map of the external Hellenides in NW Greece illustrating the principal tectonostratigraphic zones: Pindos, Gavrovo, Ionian, and Pre-Apulian Zones (modified from [12]). Red boxes show the studied areas of Kastos Island, in the external Ionian zone, and Araxos peninsula (NW Peloponnesus), in the internal Ionian zone.

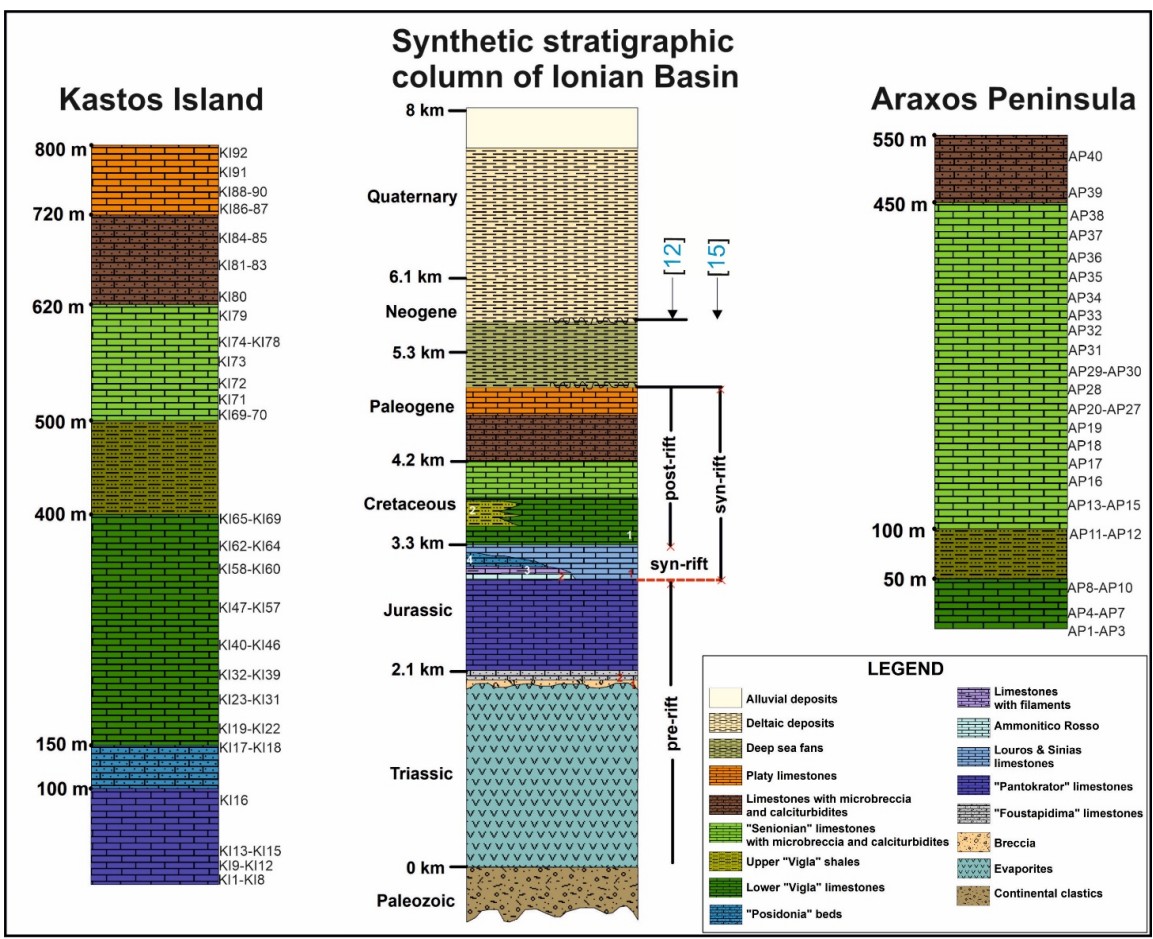

**Figure 2.** Detailed synthetic lithostratigraphic columns of Cretaceous deposits, from NW Peloponnesus and Kastos Island, in relation with the synthetic stratigraphic column of Ionian zone (modified from [12]).

The Lower Cretaceous (Berriasian to Turonian) is characterized by "Vigla limestones" and the laterally equivalent "Vigla shales," 200–600 m in total thickness. The "Vigla" formation locally overlies the pre-rift sequence and thickness variations indicate persistent differential subsidence. "Vigla limestones" consist of thin-bedded grey mudstone to wackestone in rhythmic alternations with chert intervals and rare shale intercalations "Vigla shales" consist of limestones with chert intervals and dark grey to green or red shale intercalations. According to [18], the "Vigla formation" was studied in the three parts of the Ionian basin independently for the two different units. In detail, "Vigla" limestones in the external parts of the basin consist of white, light grey to yellowish micrites, and radiolarian biomicrites (wackestones to packstones and rarely mudstones). In the middle part of the basin, it consists of yellow to red marly limestones or shaly limestones and chert alternations, as well as clay layers that are usually green and red. The calcareous beds consist mainly of micrites (mudstones, wackestones), biomicrites with foraminifera and radiolaria (foraminiferal or radiolarian wackestones to packstones), and siliceous biomicrites (Si-wackestones, packstones). In the internal parts of the basin, it consists of compact, thick-bedded, bituminous, dolomitic limestones, locally weathered, with lenses of slightly dolomitized microbreccia and thin cherty layers, chert intercalations, and nodules. The non-dolomitized beds are micritic limestones with and without radiolaria and microbreccia with thick chert intercalations. "Vigla" shales in the external parts of the basin have a micritic matrix, consist of floatstones, rudstones, and rare grainstones and packstones with micritic or biomicritic intercalations which are locally numerous. Occasionally, chert nodules and yellowish layers are also observed. In the middle part of the basin the limestones are microclastic, bioclastic, or microbreccia with a micritic matrix (wackestones, packstones,

floatstones, rudstones), intercalated with micrites and biomicrites. Chert layers are rarely observed. The internal parts of the basin are characterized by massive, thick-bedded microbreccia to breccia containing rudists and coral fragments. Rarely, layers of the platy to thick-bedded micritic to biomicritic limestones (wackestones, packstones) are intercalated. Nodules and thin layers of chert are observed locally. Over the latter, the Upper Cretaceous "Senonian limestones" (Coniacian to Maastrichtian) were deposited. They range from 200 to 400 m in thickness and contain both nodular and bedded chert. The Upper Cretaceous deposits, mentioned as "clastic limestones" ("Senonian limestones") and expanding from the western to the eastern part of the basin [18], and practically from the external to the internal Ionian zone, vary throughout the basin: A. in the external (western) parts of the basin they have a micritic matrix, consisting of floatstones, rudstones, and rarely grainstones and packstones with micritic or biomicritic intercalations. Occasionally, chert nodules and yellowish cherty layers are also observed. B. in the middle part of the basin, the limestones are microclastic, bioclastic, or contain microbreccia with a micritic matrix (wackestones, packstones, floatstones, and rudstones), intercalated with micrites and biomicrites. Chert layers are rarely observed. C. in the internal (eastern) part, limestones are massive, and thick-bedded with microbreccia to breccia containing rudist and coral fragments. Rarely, layers of platy to thick-bedded micritic to biomicritic limestones are intercalated, whereas nodules and thin layers of chert are observed locally.

The Paleocene rocks have similar lithofacies as the Upper Cretaceous "Senonian limestones," with prominent microbreccia that derived from the erosion of the Cretaceous carbonates from both the Gavrovo platform (to the east) and the Apulian platform (to the west), known as "limestones with microbreccia". At some horizons, bedded or nodular chert is also found.

The Lower Eocene rocks comprise "platy limestones" and platy wackestones/mudstones with Globigerinidae and nodular cherts, especially in the central area of the Ionian basin. These deposits look similar to the "Vigla limestones" but they lack bedded cherts.

Within the Ionian Zone, the Upper Cretaceous-Eocene resedimented carbonates are considered as one of the main reservoir sequences and exploration targets in western Greece [12,14]. They correspond to the producing reservoir rocks in the Katalokon oil field and host proved prolific reservoirs in the Ionian zone of Albania [19] and in the central and southern Adriatic offshore Italy [11,20]. However, the distribution of these resedimented carbonates (calcareous turbidites and coarser breccia) in western Greece is still poorly constrained.

Sedimentation within the Ionian basin in Albania was controlled by the tectonic instabilities of both shelf margins (Sazani and Kruja platforms). During the Cretaceous period, the deposition in the Ionian basin was characterized by hemipelagites, at the base, and by the overlying breccia and calciturbidites, produced from the eroded margins, due to their strong tectonic instability [12,21,22].

Other examples can be taken from the other side of the Adriatic Sea, the Apulian platform with its margins and slope located at the Gargano peninsula and Maiella Mountain in Italy [11,23–25]. During Cretaceous to Lower Eocene, steep submarine escarpments characterized the Apulian carbonate platform margins presenting significant erosion. Breccia accumulated at the toe of the escarpment interbedded with clinostratified deposits and bioclastic turbidites (around the Gargano promontory and Maiella Mountain, (Morrone di Pacento, Cima delle Murelle, the Grotte and Orfento formations in Maiella Platform)). Along the platform margins, significant rudist colonies existed. These colonies were subsequently eroded and were redeposited as skeletal material at the toe of the slope [25,26].

### 2.2. Studied Areas

After our previous studies [15–17] which showed different depositional conditions either due to their different tectonic activity or due to the different influence of the neighboring existing platforms (Apulian to the west and Gavrovo to the east), the same two areas were selected for the detailed study of this work. The above differences referred to

different mineralogy of siliceous concretions, different kinds of soft-sediment deformations, and different influences of tectonic activity (normal or transfer faults) in Ionian basin depositional conditions.

Kastos Island is located in the Ionian Sea and it belongs to the Ionian zone, specifically to the external Ionian zone (Figure 3). The external Ionian zone and the Pre-Apulian zone to the west form the outer continental margin of the Apulian Platform in the Ionian and Adriatic Seas [25].

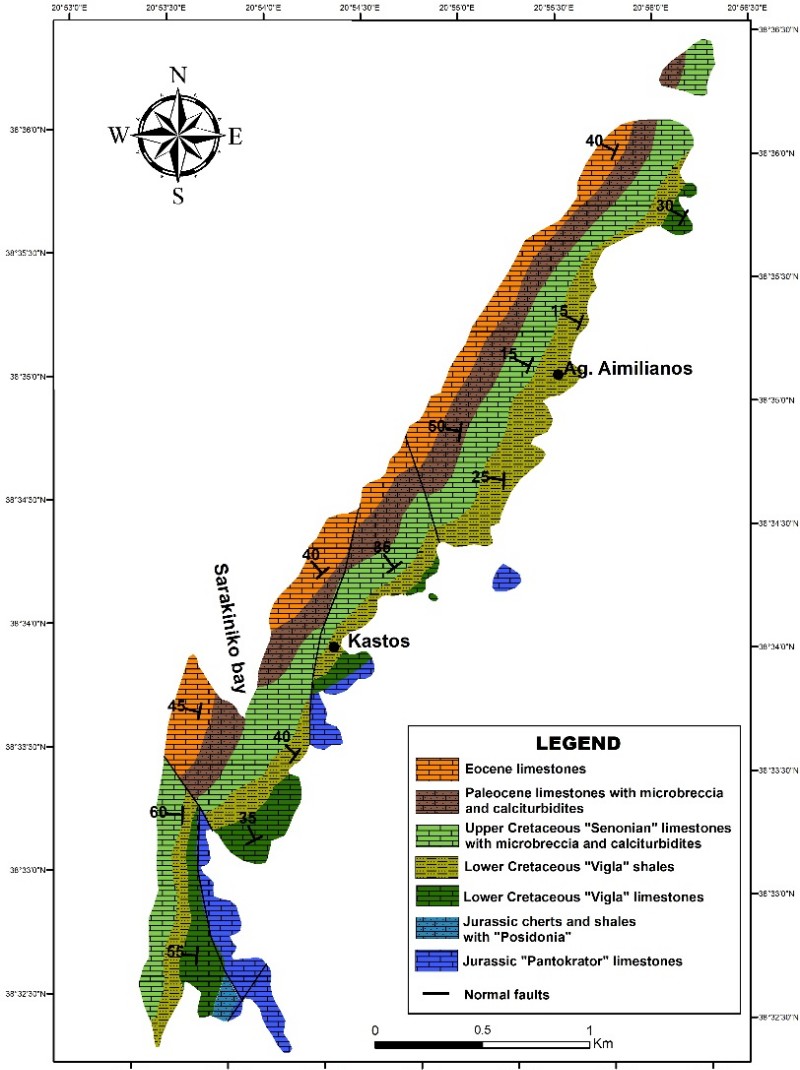

**Figure 3.** Digitized geological map of Kastos Island [16] based on an original pre-existing geological map [27].

On Kastos Island and according to the geological map of IGME sheet Kalamos [27] Jurassic to Eocene deposits crop out, nevertheless, the Mesozoic sedimentary sequence is incomplete. The oldest deposits, the Lower Jurassic "Pantokrator" limestones, crop out at the southeastern side of Kastos Island, whereas the younger Eocene carbonates are visible on the western side. There is only one site at the southern part of the island where Jurassic shales with a "Posidonia" outcrop. In all other sites, there is no evidence for the presence of Jurassic shales in the transition from Lower Jurassic "Pantokrator" limestones to Lower Cretaceous "Vigla" limestones. In general, dip direction is to the west with a dip up to 45°, and only in the southwestern part dip direction is to the south and dip up to 60°.

Araxos peninsula is located in NW Peloponnesus (Figure 4) and it belongs to the internal Ionian zone. In more detail, Araxos peninsula is situated at the western end of

Patraikos Gulf, and during Late Pliocene to present, it was affected by the tectonic regime that formed the WNW trending extensional Patras-Corinth Basin. Thus, its formation was controlled by back-arc extension behind the Hellenic trench and is characterized by WNW–ESE normal faults and NNE–SSW transfer faults [28–31]. Western Greece has been affected by rotation up to 50° clockwise (-cw) [32–36], which took place in two stages/episodes of about 25° each: the first one occurred before the Late Miocene and the second one during the Plio-Pleistocene.

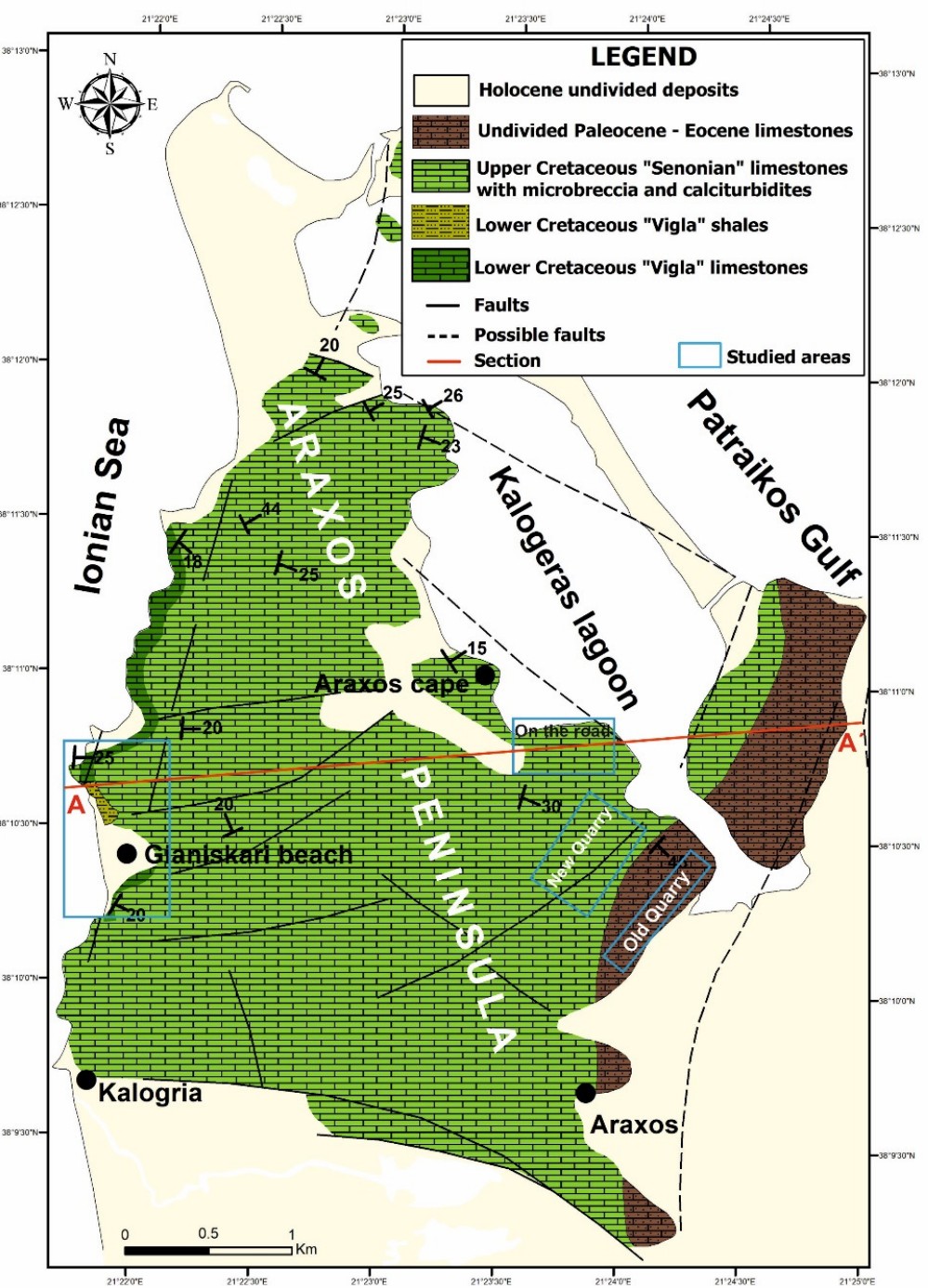

**Figure 4.** Geological map of the studied area in the Araxos peninsula (NW Peloponnesus) [15] modified from [37]. Cross-section A-A' corresponds to Figure 10, whereas blue boxes show the studied areas from where selected samples were studied in detail.

## 3. Materials and Methods

The studied material consists of ninety-two (92) samples collected from Lower Jurassic to Eocene carbonates of Kastos Island and forty (40) samples collected from the Cretaceous carbonate rocks of Araxos peninsula (Figure S1). The selection of samples was based on lithology and facies alternations. All samples were cut and thin sections were prepared. The latter were studied for microfacies analysis in order to establish the temporal and spatial changes and the evolution of the respective depositional paleoenvironment in the two studied areas. The textural characters of microfacies types were defined according to Dunham's [7] classification modified by Embry and Klovan [38] and Flügel [5] and their description includes biogenic and inorganic dominant components [39]. In addition, foraminiferal assemblages provide considerable support to the interpretation of depth and depositional environments [40,41] as well as providing biostratigraphic and consequently chronostratigraphic data, in order to define the chronological context of the studied sequences. Hence, benthic and mainly planktonic foraminifera, and where applicable calpionellids, were determined, and their stratigraphic ranges were considered to determine the age of the studied sedimentary sequences [42].

As Kastos Island was undoubtedly subjected to a compressional regime, it seems strange to present generally no deformation. Similarly, Araxos peninsula was influenced by a thrust fault situated west of the studied area [15–17], during a compressional regime. Based on these facts, the first sampling was focused on the age determination and facies analysis in specific areas as previously published works have highlighted. After the first results, additional sampling focused on the transitional zones between different age formations in order to highlight the depositional conditions' changes.

## 4. Results

### 4.1. Kastos Island

#### 4.1.1. Facies Analysis and Zones

Based on geographic and stratigraphic criteria, the studied area is organized into six specific regions: "North Kastos Island" where Lower Cretaceous to Paleocene formations are outcropping, "West and Central Kastos Island" with Lower to Upper Cretaceous outcrops, "Kastos Village" with Jurassic to Upper Cretaceous outcrops, "Sarakiniko Bay" with Lower Cretaceous to Upper Eocene outcrops, and "South Kastos Island" with Lower Jurassic to Lower Cretaceous outcrops.

The main textural and compositional characteristics of the studied total thin sections, as well as the sedimentary features of the distinguished microfacies are presented in Table 1, corresponding to different depositional environments or standard facies zones [5].

**Table 1.** Description of microfacies analysis of samples from Kastos Island and their age determination. Note the difference between the pre-existing age determinations and the new data of the present study.

| Samples | Description of Facies Analysis | Facies Zones | Age Based on Geological Map | Fossils | Age Based on Fossils |
|---|---|---|---|---|---|
| KI92 | Mudstone/wackestone (mud to grain supported), sparse micrites, skeletan grains (SMF3) | Deep sea (FZ1) | Paleocene | Radiolaria, Bivalve filaments, *Globigerinotheka sp.*, *Turborotalia sp.*, *Turborotalia cerroazulensis*, *Catapsydrax* cf. *dissimilis*, *Pseudohastigerina micra*, *Globigerina sp.* | Eocene (Bartonian-Priabonian) |
| KI91 | Grained supported, wackestone/packstone, biomicrites, skeletan grains (SMF3) | Deep sea (FZ1) | Paleocene | Radiolaria, *Globigerinotheka sp.*, *Catapsydrax sp.*, *Acarinina sp.* | middle Eocene |
| KI90 | Mudstone/wackestone, biomicrites, skeletan grains (SMF3) | Deep sea (FZ1) | Paleocene | Radiolaria, *Globigerinotheka sp.*, *Acarinina soldadoensis*, *Subbotina inaequispira* | Eocene (Lutetian) |
| KI89 | Grained supported, packstone, biosparite, peloids, calcite vein, peloids, plaktonic foraminifera, microcrystalline calcite (SMF3) | Toe of Slope (FZ3) | Paleocene | *Globigerinotheka sp.*, *Turborotalia centralis*, *Acarinina coalingensis*, *Globigerina sp.*, *Discocyclina sp.* | Eocene (Lutetian) |

**Table 1.** *Cont.*

| Samples | Description of Facies Analysis | Facies Zones | Age Based on Geological Map | Fossils | Age Based on Fossils |
|---|---|---|---|---|---|
| KI88 | Grained supported, packstone/grainstone, biosparite, calcite vein, plaktonic foraminifera, microcrystalline (SMF3) | Toe of Slope (FZ3) | Paleocene | *Globigerinotheka sp., Turborotalia centralis, Acarinina primitiva, Discocyclina sp., Assilina sp., Lepidocyclina sp., Lockhartia sp.*, Rotaliidae | Eocene (lower Lutetian) |
| KI87 | Wackestone/packstone, biomicrite, skeletal grains, calcite vein (SMF3) | Toe of Slope (FZ3) | Paleocene | *Morozovella aequa, Acarinina coalingensis, Acarinina bulbrooki, Acarinina* cf. *esnaensis* | Eocene (Ypresian) |
| KI86 | On the top: packstone with biomicrites and the other section with lamination of mudstone-wackestone (SMF3) | Deep sea (FZ1) | Paleocene | *Morozovella formosa, Igorina sp., Globalomalina planoconica, Acarinina bulbrooki* | Eocene (Ypresian) |
| KI85 | Mudstone with calcite veins (SMF1) | Deep sea (FZ1) | Paleocene | Radiolaria, *Morozovella sp., Acarinina sp.* | Paleocene |
| KI84 | Wackestone with sparse biomicrite skeletan grains, foraminifera, calcite veins (SMF3) | Deep sea (FZ1) | Paleocene | *Morozovella angulata, Morozovella aequa, Morozovella conicotruncana, Planorotalites chapmani, Subbotina inaequispira* | Paleocene (Selandian-Thanetian) |
| KI83 | Thinly laminated alternation of mudstone to wackestone (SMF3) | Deep sea (FZ1) | Senonian | *Morozovella angulata, Subbotina velascoensis, Parasubbotina varianta* | Paleocene (Selandian-Thanetian) |
| KI82 | Mudstone/wackestone, peloids or oxides, calcite veins (SMF3) | Deep sea (FZ1) | Paleocene | *Morozovella angulata, Planorotalites pseudomenardi* | Paleocene (Selandian-Thanetian) |
| KI81 | Mudstone (SMF3) | Deep sea (FZ1) | Paleocene | Radiolaria, *Morozovella angulata, Planorotalites sp.* | Paleocene (Selandian-Thanetian) |
| KI80 | Packstone microbrecciated, few skeletan grains, microsparite (SMF4) | Slope (FZ4) | Senonian | Miliolidae, Bivalve fragments, *Parasubbotina pseudobulloides* | Paleocene (Danian-Selandian) |
| KI79 | Packstone, microbrecciated (SMF4) | Toe of slope (FZ3) | Eocene | Miliolidae, *Globotruncana ventricosa, Globotruncana cf orientalis* | Late Cretaceous (Campanian-Maastrichtian) |
| KI78 | Grainstone, microbreccia (SMF5) | Slope (FZ4) | Senonian | *Helvetotruncana helvetica* | Late Cretaceous, Senonian (Turonian) |
| KI77 | Grainstone, microbrecciated, microcrystalline? (SMF2) | Deep shelf (FZ2) | Senonian | | Late Cretaceous (Senonian) |
| KI76 | Mudstone (SMF3) | Deep sea (FZ1) | Paleocne-Eocene? | Radiolaria, *Rotalipora cushmani, Clavihedbergella simplex, Hedbergella planispira* | Late Cretaceous (Cenomanian) |
| KI75 | Mudstone (SMF3) | Deep sea (FZ1) | Jurassic | Radiolaria | Late Cretaceous |
| KI74 | Mudstone (SMF3) | Deep sea (FZ1) | Jurassic | Radiolaria | Late Cretaceous |
| KI73 | chert (SMF3) | Deep sea (FZ1) | Jurassic | Radiolaria, *Clypeina jurassica* | Late Cretaceous (Kimmeridian-Tithonian) |
| KI72 | Wackestone/packstone, microbrecciated, skeletan grains (SMF4) | Toe of Slope (FZ3) | Paleocene | Radiolaria, *Rugoglobigerina sp., Globotruncana* cf. *arca, Globotruncana neotricarinata, Globotruncana orientalis, Globotruncanita cf conica* | Late Cretaceous (Maastrichtian) |
| KI71 | Grainstone, microbrecciated | Toe of Slope (FZ3) | Paleocene | | Late Cretaceous |
| KI70 | Wackestone, skeletal graines, micrites (SMF3) | Deep sea (FZ1) | Senonian | Radiolaria, *Clavihedbrgella simplex, Hedbergella planispira, Rotalipora cushmani, Praeglobotruncana delrioensis, Thalmanninella appenninica, Thalmanninella greenhornensis, Thalmanninella globotruncanoides, Whiteinella archaeocretacea* | Late Cretaceous (Cenomanian) |
| KI69 | Mudstone (SMF3) | Deep sea (FZ1) | Jurassic | Radiolaria | Late Cretaceous |

**Table 1.** *Cont.*

| Samples | Description of Facies Analysis | Facies Zones | Age Based on Geological Map | Fossils | Age Based on Fossils |
|---|---|---|---|---|---|
| KI68 | Packstone, microbioclastic, microcrystalline (SMF2) | Deep sea/deep shelf (FZ1-2) | Senonian | *Blefuscuiana gorbachikae, Blefuscuiana occulta, Blefuscuiana praetrocoidea, Globigerinelloides* cf. *ferreolensis, Globigerinelloides gottisi, Globigerinelloides barri, Globigerinelloides blowi* | Early Cretaceous (Aptian) |
| KI67 | Wackestone, microbioclastic (SMF2) | Deep sea (FZ1) | Senonian | Radiolaria, *Planomalina cheniurensis, Globigerinelloides sp., Blowiella blowi, Blowiella gottisi, Hedbergella sp.* | Early Cretaceous (Aptian) |
| KI66 | Packstone, microbioclastic, microcrystalline (SMF2) | Deep sea-deep shelf (FZ1-2) | Senonian | Radiolaria, *Blefuscuiana gorbachikae, Globigerinelloides barri, Globigerinelloides algerianus, Claviblowiella saundersi, Planomalina sp., Schackoina cepedai* | Early Cretaceous (Aptian) |
| KI65 | Mudstone with micriticous fossiliferous and calcite veins, stylolites (SMF3) | Deep sea (FZ1) | Vigla | Radiolaria, *Blowiella gottisi, Blowiella blowi, Hedbergella delrionensis, Globigerinelloides aptiensis, ?Lenticulina sp.* | Early Cretaceous (Aptian) |
| KI64 | Mudstone with micriticous fossiliferous (SMF3) | Deep sea (FZ1) | Vigla | Radiolaria, *Blowiella gottisi, Blowiella blowi, Hedbergella delrionensis, Globigerinelloides sp., Hedbergella trocoidea, Hedbergella sp., Gorbachikella cf kugleri* | Early Cretaceous (Aptian) |
| KI63 | Wackestone/Packstone, microbioclastic, microcrystalline, lamination (SMF3) | Deep sea (FZ1) | Senonian | Radiolaria, *Ticinella* cf. *praeticinencis, Hedbergella* cf. *planispira* | Early Cretaceous (Albian) |
| KI62 | Mudstone/wackestone, microbioclastic (SMF1) | Toe of Slope (FZ3) | Senonian | Radiolaria, *Hedbergella rischi, Hedbergella planispira, Hedbergella trocoidea, Hedbergella delrioensis, Hedbergella* cf. *kuznetsovae, Clavihedbergella simplex, Ticinella roberti, Alanlordella bentonensis* | Early Cretaceous (Albian) |
| KI61 | Mudstone with micriticous fossiliferous and calcite veins, stylolites (SMF3) | Deep sea (FZ1) | Vigla | Radiolaria, *Blowiella gottisi, Hedbergella delrionensis* | Early Cretaceous (Albian) |
| KI60 | A. Mudstone with micrite skeletan grains (SMF3), B. Packstone, microbioclastic, microcrystalline, lamination (SMF2) | A. Deep sea (FZ1), B. Deep sea (FZ1) | Paleocene | Radiolaria, *Hedbergella rischi* | Early Cretaceous (Albian) |
| KI59 | Mudstone-Wackestone, skeletal grains, fossiliferous biomicrite (SMF3) | Deep sea (FZ1) | Senonian | Radiolaria (Spumellaria, Nasselaria) | Early Cretaceous |
| KI58 | Packstone microbrecciated, few skeletan grains, microsparite (SMF4) | Slope (FZ4) | Senonian | | |
| KI57 | Packstone microbrecciated, few skeletan grains, microsparite (SMF4) | Slope (FZ4) | Senonian | Miliolidae, *Triloculina sp.* | |
| KI56 | Packstone/grainstone, microbrecciated, few skeletan grains, microsparite (SMF4) | Slope (FZ4) | Senonian | Benthic foraminifera | |
| KI55 | Packstone/grainstone, microbrecciated, few skeletan grains, microsparite (SMF4) | Slope (FZ4) | Senonian | Miliolidae, *Lenticulina sp.* | |

**Table 1.** *Cont.*

| Samples | Description of Facies Analysis | Facies Zones | Age Based on Geological Map | Fossils | Age Based on Fossils |
|---------|-------------------------------|--------------|---------------------------|---------|---------------------|
| KI54 | Mudstone to packstone, microcrystalline, fossiliferous biomicrite (SMF2) | Deep sea (FZ1) | Eocene | Calpionellidae, *Lorenziella* cf. *hungarica*, *Tintinnopsella* cf. *carpathica*, *Clavihedbergella eocretacea*, *Praehedbergella sigali* | Early Cretaceous (Valanginian) |
| KI53 | A. Mudstone/wackestone, skeletan grains, fossiliferous biomicrite, siliceous veins (SMF3), B. a zone which is Wackestone-Packstone, microlithoclastic, spiculite (SMF1-2) | A. Deep sea (FZ1) B. Deep sea (FZ1) | Eocene | Radiolaria, Calpionellidae, *Lorenziella* cf. *hungarica Clavihedbergella eocretacea Praehedbergella sigali Conoglobigerina sp. Lenticulina sp.* | Early Cretaceous (Valanginian) |
| KI52 | Mudstone/wackestone, skeletal grains, fossiliferous biomicrite (SMF3) | Deep sea (FZ1) | Eocene | Radiolaria, Calpionellidae, *Lorenziella* cf. *hungarica, Tintinnopsella* cf. *carpathica, Calpionellites* cf. *darderi, Clavihedbergella eocretacea Praehedbergella sigali* | Early Cretaceous (Valanginian) |
| KI51 | A. Mudstone to wackestone, microbreccia (SMF2), B. Packstone to grainstone, stylolites (SMF2) | A. Deep sea (FZ1), B. Deep sea (FZ1) | Paleocene | Radiolaria, *Planomalina cherouniensis, Blowiella blowi, Blowiella gottisi* | Early Cretaceus (Aptian) |
| KI50 | Wackestone/packstone, microbreccia (SMF2) | Deep sea (FZ1) | Paleocene | Radiolaria, *Blefuscuianna aptianna, Praehedbergella* cf. *sigali, Globigerinelloides sp, Blowiella blowi, Blowiella gottisi* | Early Cretaceus (Aptian) |
| KI49 | Wackestone and locally packstone, microbreccia (SMF2) | Deep sea (FZ1) | Paleocene | *Blowiella sp.* | Early Cretaceus (Aptian) |
| KI48 | Wackestone, microbreccia, lamination, grainstone flows, stylolites (SMF2) | Deep sea (FZ1) | Paleocene | Radiolaria, *Hedbergella planispira, Hedbergella delrioensis, Praehedbergella sp., Globigerinelloides algerianus, Blowiella duboisi, Blowiella sp.* | Early Cretaceus (Aptian) |
| KI47 | Wackestone/packstone and locally grainstone, microbreccia, lamination (SMF2) | Deep sea (FZ1) | Paleocene | *Blowiella sp., Globigerinelloides sp.* | Early Cretaceus (Aptian) |
| KI46 | Wackestone, (crystallization), skeletan grains (SMF3) | Deep sea (FZ1) | Vigla | Radiolaria | Early Cretaceous |
| KI45 | Wackestone/packstone, micrite skeletan grains (SMF3) | Deep sea (FZ1) | Vigla | Radiolaria | Early Cretaceous |
| KI44 | Mudstone, quite cracked (SMF3) | Deep sea (FZ1) | Vigla | Radiolaria | Early Cretaceous |
| KI43 | Wackestone, micrite skeletan grains, calcite vein (SMF3) | Deep sea (FZ1) | Vigla | Radiolaria | Early Cretaceous |
| KI42 | Mudstone/wackestone, skeletan grains, calcite vein, biomicrite (SMF3) | Deep sea (FZ1) | Vigla | Radiolaria | Early Cretaceous |
| KI41 | Mudstone, biomicrite (SMF3) | Deep sea (FZ1) | Senonian | Radiolaria, *Blowiella maridalensis, Hedbergella delrionensis, Globigerinelloides algerianus, Globigerinelloides ferreolensis, Blefuscuianna aptiana* | Early Cretaceus (Aptian) |
| KI40 | Mudstone, lamination? (SMF3) | Deep sea (FZ1) | Senonian | Radiolaria, *Hedbergella rischi, Hedbergella delrioensis, Hedbergella planispira, Shackoina sp.* | Early Cretaceous (Albian) |
| KI39 | Mudstone/wackestone, skeletan grains biomicrite, Stylolites (SMF3) | Deep sea (FZ1) | Vigla | Radiolaria, A few bivalve filaments, Calpionellidae, *Tintinnopsella cf longovalata, Tintinnopsella cf carpathica* | Early Cretaceous (Valanginian) |
| KI38 | Mudstone, skeletan grains, biomicrite (SMF3) | Deep sea (FZ1) | Vigla | Radiolaria, A few bivalve filaments, *Favusella sp., Blefuscuianna cf aptianna, Blowiella sp., Schackoina cepedai* | Early Cretaceous (Aptian) |

**Table 1.** *Cont.*

| Samples | Description of Facies Analysis | Facies Zones | Age Based on Geological Map | Fossils | Age Based on Fossils |
|---------|-------------------------------|--------------|---------------------------|---------|---------------------|
| KI37 | Mudstone, skeletal grains, fossiliferous biomicrite (SMF3) | Deep sea (FZ1) | Vigla | Radiolaria, *Hedbergella planispira*, *Alanlordella bentonensis* | Early Cretaceous (Albian) |
| KI36 | Mudstone, skeletan grains (SMF3) | Deep sea (FZ1) | Vigla | Radiolaria, *Hedbergella rischi*, *Hedbergella cf trocoidea*, *Planomalina sp.*, *Clavihedbrgella simplex* | Early Cretaceous (Albian) |
| KI35 | Mudstone (SMF3) | Deep sea (FZ1) | Vigla | Radiolaria, *Favusella hoterivica*, *Blowiella sp.* | Early Cretaceous (Berriasian-Aptian) |
| KI34 | Mudstone/Wackestone, skeletal grains, fossiliferous biomicrite, stylolites (SMF3) | Deep sea (FZ1) | Vigla | Radiolaria Calpionellidae, *Tintinnopsella sp.*, *Tintinnopsella* cf. *carpathica*, *Calpionellites* cf. *coronatus* | Early Cretaceous (Valanginian) |
| KI33 | Mudstone/Wackestone, skeletal grains, fossiliferous biomicrite (SMF3) | Deep sea (FZ1) | Vigla | Radiolaria Calpionellidae, *Tintinnopsella* cf. *Longovalata*, *Tintinnopsella* cf. *carpathica* | Early Cretaceous (Berriasian- Valanginian) |
| KI32 | Mudstone/Wackestone, skeletal grains, fossiliferous biomicrite (SMF3) | Deep sea (FZ1) | Vigla | Radiolaria Calpionellidae, *Tintinnopsella sp.* | Early Cretaceous (Tithonian- Valanginian) |
| KI31 | Mudstone/Wackestone, skeletal grains, fossiliferous biomicrite (SMF3) | Deep sea (FZ1) | Vigla | Radiolaria, *cf Favusella hoterivica* | Early Cretaceous (Berriasian- Barremian) |
| KI30 | Mudstone/Wackestone, skeletan graines, micrites, calcite vein (SMF3) | Deep sea (FZ1) | Vigla | *?Conoglobigerina sp.* | Early Cretaceous (Berriasian- Valanginian) |
| KI29 | Mudstone and microspar, microcrystalline (SMF3) | Deep sea (FZ1) | Paleocene-Eocene? | | Early Cretaceous |
| KI28 | Mudstone (SMF3) | Deep sea (FZ1) | Paleocne-Eocene? | Radiolaria, *Shackoina cepedai*, *Blefuscuiana aptiana*, *Hedbergella delrionensis* | Early Cretaceous (Aptian) |
| KI27 | Mudstone (SMF3) | Deep sea (FZ1) | Paleocne-Eocene? | Radiolaria, *Blowiella blowi*, *Praehedbergella cf sigali*, *Hedbergella similis* | Early Cretaceous (Aptian) |
| KI26 | Wackestone/mudstone, locally lamination (SMF3) | Deep sea (FZ1) | Paleocne-Eocene? | Radiolaria, *Blowiella gottisi*, *Biglobigerinella barri*, *Shackoina cepedai*, *Blefuscuiana aptiana*, *Blefuscuiana kuznetsovae*, *Planomalina cheniourensis*, *Globigerinelloides sp.* | Early Cretaceous (Aptian) |
| KI25 | Mudstone (SMF3) | Deep sea (FZ1) | Vigla | Radiolaria, *Blefuscuiana sp.* | Early Cretaceous (Barremian-Albian) |
| KI24 | Mudstone (SMF3) | Deep sea (FZ1) | Jurassic | Radiolaria | Early Cretaceous |
| KI23 | Mudstone (SMF3) | Deep sea (FZ1) | Jurassic | Radiolaria, Calpionellidae, *Tintinnopsella sp.* | Early Cretaceous (Tithonian- Valanginian) |
| KI22 | Mudstone (SMF3) | Deep sea (FZ1) | Jurassic | Radiolaria, *Blowiella sp.*, *Blefuscuianna kuznetsovae* | Early Cretaceous (Aptian) |
| KI21 | Mudstone (SMF3) | Deep sea (FZ1) | Vigla | Radiolaria | Early Cretaceous |
| KI20 | Mudstone (SMF3) | Deep sea (FZ1) | Jurassic | Radiolaria | Early Cretaceous |
| KI19 | Mudstone, stylolites (SMF3) | Deep sea (FZ1) | Jurassic | Radiolaria, *Favusella hoterivica* | Early Cretaceous (Berriasian- Barremian) |
| KI18 | Mudstone/wackestone (SMF3) | Deep sea (FZ1) | Vigla | Radiolaria, *Blowiella sp.*, *Hedbergella derlionensis*, *Hedbergella trocoidea* | Early Cretaceous |
| KI17 | Mudstone (SMF3) | Deep sea (FZ1) | Vigla | Radiolaria | |
| KI16 | Mudstone, stylolites (SMF3) | Deep sea (FZ1) | Jurassic | Radiolaria, a few Bivalve filaments Calpionellidae, *Calpionellites sp.*, *Tintinnopsella sp.* | Early Cretaceous (Valanginian) |
| KI15 | Wackestone/grainstone and sometimes boundstone (SMF18) | Open marine/restricted (FZ7-8) | Jurassic | *Siphovalvulina sp.* | Early Jurassic |
| KI14 | Wackeston/grainstone and sometimes boundstone (SMF18) | Open marine/restricted (FZ7-8) | Jurassic | *Siphovalvulina sp.*, *Duotaxis sp.* | Early Jurassic |
| KI13 | Wackeston/grainstone and sometimes boundstone (SMF18) | Open marine/restricted (FZ7-8) | Jurassic | | Early Jurassic |

**Table 1.** *Cont.*

| Samples | Description of Facies Analysis | Facies Zones | Age Based on Geological Map | Fossils | Age Based on Fossils |
|---|---|---|---|---|---|
| KI12 | Grainstone (SMF18) | Open marine/restricted (FZ7-8) | Jurassic | ?*Coronipora sp.* | Early Jurassic |
| KI11 | Grainstone (SMF18) | Open marine/restricted (FZ7-8) | Jurassic | *Siphovalvulina sp., Duotaxis sp., Triloculina sp.* | Early Jurassic |
| KI10 | Wackestone/grainstone and sometimes boundstone (SMF18) | Open marine-restricted (FZ7-8) | Jurassic | *Thaumatoporella sp., Siphovalvulina sp.* | Early Jurassic |
| KI9 | Grainstone and sometimes boundstone (SMF18) | Open marine/restricted (FZ7-8) | Jurassic | *Thaumatoporella sp., Siphovalvulina sp.* | Early Jurassic |
| KI8 | Grainstone and sometimes boundstone, locally pelsparite (SMF18-19) | Open marine/restricted (FZ7-8) | Jurassic | *Thaumatoporella sp.* | Early Jurassic |
| KI7 | Grainstone and sometimes boundstone (SMF18) | Open marine/restricted (FZ7-8) | Jurassic | *Thaumatoporella sp., Siphovalvulina sp., Triloculina sp.*, Miliolidae | Early Jurassic |
| KI6 | Wackestone/grainstone and sometimes boundstone (SMF18) | Open marine/restricted (FZ7-8) | Jurassic | *Thaumatoporella sp., Siphovalvulina sp.* | Early Jurassic |
| KI5 | Wackestone/grainstone and locally boundstone (SMF18-19) | Open marine/restricted (FZ7-8) | Jurassic | Thaumatoporella sp., Siphovalvulina sp., Miliolidae | Early Jurassic |
| KI4 | Wackestone/grainstone (SMF18-19) | Open marine/restricted (FZ7-8) | Jurassic | *Thaumatoporella sp., Siphovalvulina sp.* | Early Jurassic |
| KI3 | A. Grainstone with peloids (pelsparite) (SMF18), B. Mudstone with lamination (SMF2), stylolites, Facies A probably an exoclast from pantokrator formation limestone | A. Open marine/restricted (FZ7-8) B.Deep sea (FZ1) | Jurassic | Radiolaria, Miliolidae | Early Jurassic |
| KI2 | Grainstone/boundstone (SMF18) | Open marine (FZ7) | Jurassic | *Thaumatoporella sp., Siphovalvulina sp.*, Algae | Jurassic |
| KI1 | Wackestone/packstone (SMF3) | Toe of Slope (FZ3) | Jurassic | Bivalve filaments | Jurassic |

More specifically, the Lower Jurassic ("Pantokrator" limestones) deposits consisting of wackestone-grainstone and locally boundstone limestones (SMF18-19) are biomicrites to biosparites accumulated in an open marine-restricted environment (FZ7-8).

The Lower Cretaceous deposits (Vigla limestones) were classified as mudstone/wackestone limestones consisting of micrites to biomicrites with planktonic foraminifera and radiolaria (standard microfacies SMF1-3), indicating a deep-sea to slope environment (FZ1-4). In the Lower Cretaceous (Valanginian), the observed mudstone-wackestone limestones consist of biomicrites with skeletal grains and spiculites (SMF3, SMF1) and are associated with a deep-sea environment (FZ1).

Calciturbidites of Late Cretaceous were classified as packstones biomicrites with microbreccia and planktonic foraminifera (SMF4) in a slope environment (FZ4).

The Paleocene deposits consist of mudstone to wackestone biomicrites with skeletal grains (SMF3) and sometimes packstone biomicrites (SMF4).

In addition, the Lower Eocene deposits consist of wackestone-packstone biomicrites (SMF3) and are related to a deep-sea or toe of slope environment (FZ1 to FZ3).

4.1.2. Biostratigraphy of Kastos Island

The biostratigraphic analysis, on the selected thin sections, revealed the following assemblages (Table 1):

The age determinations (Table 1) do not fully support the pre-existing age results as they are reported in the geological map (Kalamos geological sheet) concerning the Ionian zone rock exposures on Kastos Island. Representative photographs showing textures and characteristic fossils are presented in Figure 5.

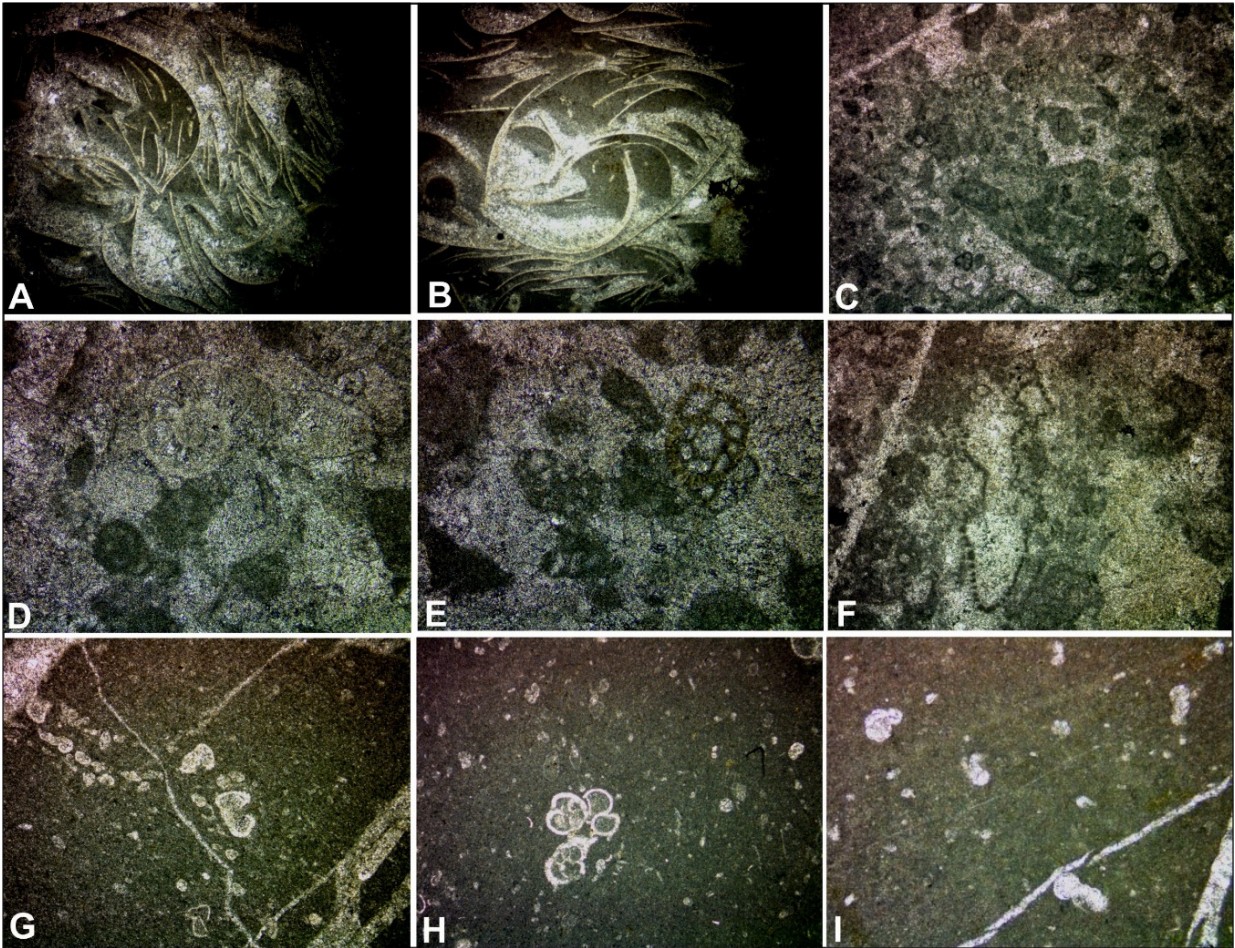

**Figure 5.** (**A**) KI27 sample of Lower Cretaceous (Valanginian): thin shelled bivalves (filaments) ($^X$20), (**B**) KI27 sample of Lower Cretaceous (Valanginian): thin shelled bivalves (filaments) ($^X$20), (**C**) KI14 sample of Lower Jurassic: small benthic foraminifera ($^X$20), (**D**) KI66 sample of Lower Cretaceous (Albian): grainstone with small benthic foraminifera ($^X$100), (**E**) KI66 sample of Lower Cretaceous (Albian): grainstone with small benthic foraminifera ($^X$100), (**F**) KI12 sample of Lower Jurassic: Thaumatoporella sp. ($^X$40), (**G**) KI83 sample of Paleocene (Selandian-Thanetian): mudstone with Morozovella angulata ($^X$40), (**H**) KI92 sample of Eocene (Bartonian-Priabonian): wackestone with Globirerinatheca sp. ($^X$40), (**I**) KI72 sample of Upper Cretaceous (Cenomanian): wackestone with Hedbergella delrioensis ($^X$100).

### *4.2. Araxos Peninsula*

#### 4.2.1. Facies Analysis and Zones

The studied area is organized into three specific areas: "Gianiskari beach" where Lower Cretaceous to Upper Cretaceous formations crop out, "on the road" with Upper Cretaceous calciturbidites and breccia-microbreccia outcrops, and "Quarry" (old and new) Upper Cretaceous to Paleocene outcrops.

In detail, the following microfacies were determined and their supported facies zones that were recognized (Table 2), are:

**Table 2.** Detailed description of studied thin sections where depositional facies, fossils, and the age of the studied deposits are presented in Araxos peninsula.

| Samples | Area | Description of Facies Analysis | Facies Zones | Age Based on Geological Map | Fossils | Age Based on Fossils |
|---|---|---|---|---|---|---|
| AP40 | Quarry | Pelagic wackestone. Most of the planktonic foraminifera tests oriented parallel to the bedding. Calcite veins. (SMF3) | Deep sea/Deep shelf (FZ1/FZ2) | Senonian | Radiolaria, *Morozovella aequa* | Paleocene (Selandian-Thanetian) |
| AP39 | Quarry | Pelagic wackestone (SMF3) | Deep shelf (FZ2) | Paleocene | Radiolaria, *Parasubbotina pseudobulloides, Acarinina sp., Subbotina sp., Igorina pussila, Chiloguemberlita sp.* | Paleocene (Selandian) |
| AP38 | Quarry | Wackestone with a "flow" of packstone with planktonic foraminifera (SMF3) | Toe of Slope (FZ3) | Senonian | Ostracods, planktonic foraminifera | Late Cretaceous (Senonian) |
| AP37 | Quarry | Calciturbidite (calcilutite), Packstone (SMF4) | Slope (FZ4) | Senonian | Radiolaria, small planktonic foraminifera | Late Cretaceous (Senonian) |
| AP36 | Quarry | Micro brecciated limestone with nodules just over decollement surface (SMF4) | Slope (FZ4) | Senonian | Radiolaria, small planktonic foraminifera | Late Cretaceous (Senonian) |
| AP35 | Gianiskari | Mudstone/wackestone (SMF3) | Deep sea (FZ1) | Senonian | Radiolaria | Late Cretaceous (Senonian) |
| AP34 | Quarry | Calciturbidite (calcarenite), Packstone with small-sized bioclasts of shoal fauna (rudists, benthic forams and echinoderms). Mudstone/wackestone (SMF4) | Slope (FZ4) | Senonian | Radiolaria, Heterohelicidae, small planktonic foraminifera | Late Cretaceous (Senonian) |
| AP33 | Quarry | Pelagic mudstone with Globotruncana (SMF3) | Deep shelf (FZ2) | Senonian | *Hedbergella sp.,* cf. *Abathomphalus mayaroensis* | Late Cretaceous (Maastrichtian) |
| AP32 | Quarry | Wackestone and calcite veins filled with coarse calcite grains. Calciturbidite (calcilutite), wackestone/packstone with small-sized bioclasts (rudists, echinoderms) (SMF4) | Slope (FZ4) | Senonian | Radiolaria, *Globotruncana arca, Globotruncanella havanensis, Rugoglobigerina sp., Heterohelix sp., ?Abathomphalus mayaroensis* | Late Cretaceous (Maastrichtian) |
| AP31 | Quarry | Calciturbidite (Calcirudite), floatstone with bioclasts and intraclasts (SMF5) | Slope (FZ4) | Senonian | Algae, Echinoderm fragments, Rudist fragments, Rudist, Miliolidae, *Textularia sp., Cuneolina sp., Quinqueloculina sp., Nezzazata sp., Alanlordella messinae, Orbitoides media, Globotruncana arca Globotruncana linneiana, Globotruncana* cf. *ventricosa Globotruncanella havanensis, Globotruncanita stuarti, Globotruncanita pettersi, Contusotruncana sp., Gannserina gannseri, Rugoglobigerina sp., Rugoglobigerina rugosa, Heterohelix sp., Abathomphalus mayaroensis* | Late Cretaceous (Maastrichtian) |
| AP30 | Quarry | Calciturbidite (calcirudite), floatstone/packstone with coarse bioclasts (SMF5) | Slope (FZ4) | Senonian | Algae, Echinoderm fragments, Rudist fragments, Miliolidae, *Cuneolina sp., Orbitoides sp., Lepidorbitoides sp., Globotruncanita stuarti, Rotalia sp., Rugoglobigerina rugosa,* Ostrascods, *Abathomphalus mayaroensis* | Late Cretaceous (Maastrichtian) |

Table 2. *Cont.*

| Samples | Area | Description of Facies Analysis | Facies Zones | Age Based on Geological Map | Fossils | Age Based on Fossils |
|---|---|---|---|---|---|---|
| AP29 | Quarry | Calciturbidite (calcirudite), floatstone with bioclasts of rudists, benthic forams from a nearby shoal (SMF5) | Slope (FZ4) | Senonian | Algae, Dacycladacae, Mollusc fragments, Rudist fragments, Miliolidae, *Cuneolina sp.*, *Orbitoides media*, *Lepidorbitoides sp.*, *Sulcoperculina sp.*, *Siderolites calcitrapoides*, *Alanlordella messinae*, *Globotruncana esnehensis*, *Globotruncana ventricosa Globotruncanita stuarti*, cf. *Contusotruncana contusa*, *Gannserina gannseri*, *Rugoglobigerina sp.*, *Abathomphalus mayaroensis* | Late Cretaceous (Maastrichtian) |
| AP28 | On the road | Biomicrite, floatstone/ packstone (SMF4) | Toe of Slope (FZ3) | Senonian | *Globotruncana arca*, *Globotruncanita stuarti*, *Rugoglobigerina rugosa*, *Abathomphalus mayaroensis*, Rudist fragments, *Orbitoides sp.* | Late Cretaceous (Maastrichtian) |
| AP27 | On the road | Microbreccia biolithoclastic packstone. Bioclasts (SMF2) | Deep shelf/Deep sea (FZ1/FZ2) | Senonian | *Globotruncanita stuarti*, *Abathomphalus mayaroensis*, *Rugoglobigerina rugosa*, Radiolaria | Late Cretaceous (Maastrichtian) |
| AP26 | On the road | Allochthonous bioclastic rudstone/ floatstone breccia, geopetal fractures (SMF5) | Slope (FZ4) | Senonian | Rudist fragments, Radiolaria, Miliolidae, Algae, *Orbitoides sp.*, *Globotruncana* cf. *bulloides* | Late Cretaceous (Campanian–Maastrichtian) |
| AP25 | Quarry | Allochthonous bio-lithoclastic packstone/rudstone, oosparite endoclast, micritic endoclast (SMF5) | Slope (FZ4) | Senonian | *Lepidorbitoides sp.*, Rudist fragments, Mollusc fragments, *Sulcoperulina sp.* | Late Cretaceous (Campanian–Maastrichtian) |
| AP24 | Quarry | Allochthonous bio-lithoclastic packstone with microsparite and micritic clasts (SMF4) | Toe of Slope (FZ3) | Senonian | Rotaliidae, *Hedbergella sp.*, *Rugoglobicerina sp.*, Miliolidae, *Gannserina gannseri*, *Globotruncana arca* | Late Cretaceous (Campanian–Maastrichtian) |
| AP23 | Quarry | Allochthonous bio-lithoclastic packstone. Coarse crystalline mosaic of calcite with micritic clasts and fenestral caves. Also the typical dog-teeth cement has developed around some breccia and suggests meteoric diagenesis (SMF5) | Slope (FZ4) | Senonian | Radiolaria, Rotaliidae, *Hedbergella sp.*, Miliolidae, *Gansserina gansseri* | Late Cretaceous (Campanian–Maastrichtian) |
| AP22 | Quarry | In places, grainstone/rudstone (higher energy). Floatstone/rudstone with bioclasts of rudists and mollusk fragments, intraclasts with benthic foraminifera. Environment with high energy (SMF5) | Slope (FZ4) | Senonian | *G* astropod fragments, Rudist fragments, Bivalve filaments, Miliolidae, *Quinqueloculina sp.*, *?Nezzazata sp.*, *Orbitoides media*, *Rugoglobigerina rugosa*, *Heterohelix sp.*, *Hedbergella sp.*, *Abathomphalus mayaroensis* | Late Cretaceous (Campanian–Maastrichtian) |
| AP21 | Quarry | In places, grainstone/rudstone (higher energy). Wackestone (SMF5) | Slope (FZ4) | Senonian | Radiolaria, *Rugoglobigerina sp.*, *Rugoglobigerina rugosa*, *Rugoglobigerina pennyi*, *Planomalina sp.*, *Hedbergella sp.* | Late Cretaceous (Campanian–Maastrichtian) |
| AP20 | Quarry | Allochthonous bio-lithoclastic packstone/rudstone, oosparite endoclast, micritic endoclast (SMF5) | Slope (FZ4) | Senonian | Mollusc fragments, Benthic foraminifera fragments | Late Cretaceous (Campanian–Maastrichtian) |

**Table 2.** *Cont.*

| Samples | Area | Description of Facies Analysis | Facies Zones | Age Based on Geological Map | Fossils | Age Based on Fossils |
|---|---|---|---|---|---|---|
| AP19 | Quarry | Allochthonous bio-lithoclastic packstone/rudstone, oosparite endoclast, micritic endoclast (SMF5) | Slope (FZ4) | Senonian | *Orbitoides sp., Orbitoides* cf. *media, Lepidorbitoides sp., Siderolites calcitrapoides, Textularia sp.,* Mollusc fragments, Algae fragment | Late Cretaceous (Campanian–Maastrichtian) |
| AP18 | On the road | Biopelmicrite with scattered benthic foraminifera and echinoderm fragments. Fenestral caves maybe to meteoric diagenesis (SMF19) | Restricted platform (FZ8) | Senonian | *Quinqueloculina sp., Spiroloculina sp., Pseudolituonella sp., Cuneolina sp.,* Textulariidae, Miliolidae | Late Cretaceous (Campanian–Maastrichtian) |
| AP17 | Quarry | Packstone/grainstone, with endoclast (SMF4) | Toe of Slope (FZ3) | Senonian | *Globotruncanita* cf. *stuartiformis, Globotruncana linneiana, Globotruncana ventricosa, Cuneolina sp.,* Rudist fragments, Mollusc fragments | Late Cretaceous (Campanian) |
| AP16 | Quarry | Microbracciated limestone with abundant bioclastic fragments and micritic clasts, ooids fragments, geopetal fabrics (SMF5) | Slope (FZ4) | Senonian | Miliolidae, Mollusc fragments, Cuneolina sp., ?*Globotruncanita stuartiformis, Rugoglobicerina sp.* | Late Cretaceous (lower Campanian) |
| AP15 | Quarry | Wackestone /Packstone with material derived from reef organisms (SMF4) | Slope (FZ4) | Senonian | Radiolaria, algae, Echinoderm fragments, Rudist fragments, Miliolidae, Rotaliidae, Textularia sp., Nodosaria sp., Orbitoides media | Late Cretaceous (Santonian–Maastrichtian) |
| AP14 | Quarry | Bioclastic Packstone (SMF4) | Slope (FZ4) | Senonian | *Alanlordella messinae, Rugoglobigerina sp.,* Radiolaria | Late Cretaceous (Santonian–Maastrichtian) |
| AP13 | On the road | Calciturbidite (calcarenite), Packstone with small-sized bioclasts of shoal fauna (rudists, benthic forams and echinoderms). Packstone/wackestone (SMF4) | Slope (FZ4) | Senonian | Radiolaria, *Rugoglobigerina sp., Marginotruncana pseudolinneiana Planomalina sp. Hedbergella sp., Globigerinelloides sp.* | Late Cretaceous (Turonian–Santonian) |
| AP12 | | Microbrecciated, two different facies separated by unconformity a. mudstone (SMF1) b. microcrystalline with calcite vein | Deep sea (FZ1) | | | Middle Cretaceous |
| AP11 | | a.Mudstone microcrystalline b. a zone which is Wackestone- Packstone, microlithoclastic (SMF3) | Deep sea/deep shelf (FZ1/2) | | | Middle Cretaceous |
| AP10 | Gianiskari beach | The same as above, wackestone with bioclast fragments (SMF3) | Deep sea (FZ1) | Vigla | Radiolaria, Algae | Early Cretaceous (Albian) |
| AP9 | Gianiskari beach | Biomicrite. Wackestone with bioclasts and ooids(transported) (SMF3) | Deep sea/deep shelf (FZ1/2) | Vigla | Radiolaria | Early Cretaceous (Albian) |
| AP8 | Gianiskari beach | Mudstone crossed by crosscutting veins (SMF3) | Deep sea (FZ1) | Vigla | Radiolaria | Early Cretaceous (Albian) |
| AP7 | Gianiskari beach | Wackestone with pelagic fauna (calcified radiolaria?) (SMF3) | Deep sea (FZ1) | Vigla | Radiolaria, *Ticinella sp* | Early Cretaceous (Valanginian) |
| AP6 | Gianiskari beach | Wackestone. Calcite veins have been obderved (SMF3) | Deep sea (FZ1) | Vigla | Radiolaria, *Ticinella sp* | Early Cretaceous (Valanginian) |
| AP5 | Gianiskari beach | Mudstone (SMF1) | Deep sea (FZ1) | | | Early Cretaceous |

**Table 2.** *Cont.*

| Samples | Area | Description of Facies Analysis | Facies Zones | Age Based on Geological Map | Fossils | Age Based on Fossils |
|---------|------|-------------------------------|--------------|---------------------------|---------|---------------------|
| AP4 | Gianiskari beach | Mudstone, skeletan grains, biomicrite (SMF3) | Deep sea (FZ1) | | | Early Cretaceous |
| AP3 | Gianiskari beach | Calciturbidite (calcarenite), Wackestone with sparse pelagic fauna (calcified radiolarian?) (SMF2) | Deep shelf (FZ2) | Vigla | Radiolaria, Algae, Calpionelidae | Early Cretaceous (Tithonian–Valanginian) |
| AP2 | Gianiskari beach | Two facies: a. mudstone-wackestone, skeletan grains, microcrystalline (SMF3), b. wackestone-packstone (SMF2) | a. deep sea (FZ1), b. deep sea/deep shelf (FZ1-2) | Vigla | Calpionellidae | Early Cretaceous (Tithonian–Valanginian) |
| AP1 | Gianiskari beach | Packstone-Wackestone, skeletal grains, fossiliferous biomicrite, calcite vein (SMF3) | Deep sea (FZ1) | Vigla | Calpionellidae | Early Cretaceous (Tithonian–Valanginian) |

More specifically, the Lower Cretaceous deposits (Vigla limestones) were classified as mudstone/wackestone limestones consisting of micrites to biomicrites with planktonic foraminifera and radiolaria (standard microfacies SMF1-3), indicating a deep-sea to slope environment (FZ1-4). In the Lower Cretaceous (Valanginian), the observed mudstone-wackestone limestones, consist of biomicrites with skeletal grains and spiculites (SMF3, SMF1) and are associated with a deep-sea environment (FZ1).

The Upper Cretaceous calciturbidites were classified as packstone biomicrites with microbreccia and planktonic foraminifera (SMF4) in a slope environment (FZ4).

The Paleocene deposits consist of grain-supported wackestone-grainestone biomicrites with skeletal grains.

### 4.2.2. Biostratigraphy of Araxos Peninsula

The age determination (Table 2) supports the pre-existing results reported in the geological map concerning the Ionian zone rock exposures sheet Nea Manolas [38]. In particular, they confirm the Lower Cretaceous age for the "Vigla" limestones, the Upper Cretaceous age for the calciturbidites and brecciated limestones, and the Paleocene age for the uppermost part of the studied carbonate sequence.

Representative photographs showing sedimentological textures and characteristic fossils are presented in Figure 6.

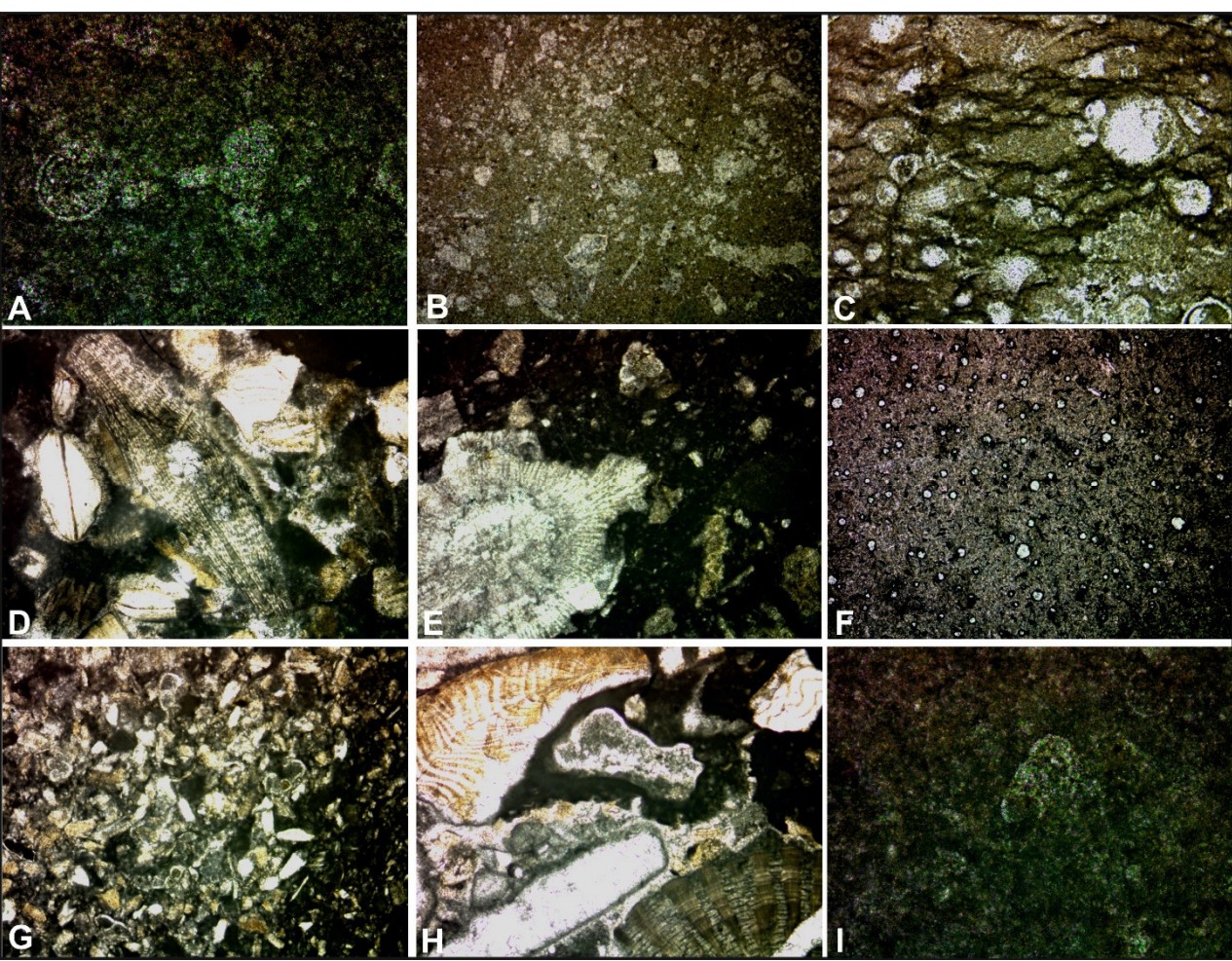

**Figure 6.** (**A**) AP29 sample of Upper Cretaceous (Maastrichtian): wackestone with Alanlordella bentonensis ($^X$400), (**B**) AP32 sample of Upper Cretaceous (Maastrichtian): wackestone with Globotruncana arca ($^X$40), (**C**) AP9 sample of Lower Cretaceous (Albian): numerus Radiolarian specimens ($^X$400), (**D**) AP19 sample of Upper Cretaceous (Campanian-Maastrichtian): packstone with Lepidorbitoides sp. ($^X$40), (**E**) AP29 sample of Upper Cretaceous (Maastrichtian): Siderolites calcitrapoides ($^X$40), (**F**) AP6 sample of Lower Cretaceous: wackestone with numerous Radiolarian tests ($^X$40), (**G**) AP27 sample of Upper Cretaceous (Maastrichtian): packstone with planktonic foraminifera (Abathomphalus mayaroensis, $^X$40), (**H**) AP26 sample of Upper Cretaceous (Campanian-Maastrichtian): Orbitoides sp., rudists fragments ($^X$40), (**I**) Blefuscuiana gorbachicae ($^X$400).

## 5. Discussion

Integrated data obtained from the microfacies analysis of the studied sections (Jurassic to Eocene deposits in Kastos Island and Early Cretaceous to Eocene in Araxos peninsula) represent an efficient interpretation tool for reconstructing temporal trends in the spatial distribution of distinct microfacies types.

By means of microfacies analysis, on Kastos Island different depositional environments were identified. In general, results show a general transition from open marine and/or restricted conditions during the Early Jurassic ("Pantokrator formation") to deep-sea conditions during the Late Jurassic (Jurassic radiolarites and shales with "Posidonia") and Early Cretaceous ("Vigla formation"). Gradually, during the Late Cretaceous ("Senonian formation"), the deep-sea conditions changed to slope conditions introducing a gradual shallowing. Paleocene and Eocene present many internal changes between deep-sea to toe of slope conditions. It seems that at the Paleocene/Eocene boundary, the shallowest conditions are represented by the toe of the slope, whereas during the Eocene, a general gradual deepening upwards to deep-sea conditions occurred.

Moreover, the deformation that was recognized within the Jurassic shales with "Posidonia" in the southern part of the island, the fact that the underlying Jurassic "Pantokrator" limestones and the overlying Lower Cretaceous "Vigla" limestones remained undeformed, and that in the same area the Jurassic "Pantokrator formation" limestones are found in contact with "Vigla" limestones without the presence of Jurassic shales support the idea that the Jurassic shales likely slumped from the top of "Pantokrator formation" just after sedimentation and moved basinwards.

From the above, it is suggested that the syn-rift stage started just after the sedimentation of the Jurassic shales. Consequently, the absence of Jurassic shales in the transitional zone from "Pantokrator" to "Vigla" formations could be plausible due to erosion.

In Araxos peninsula, and due to the fact that only the upper part of the Lower Cretaceous deposits is outcropping whereas the Jurassic deposits are absent, only a few changes in the depositional conditions were observed. The Lower Cretaceous "Vigla" limestones show changes from deep-sea to middle and outer slope, whereas the Upper Cretaceous "Senonian" deposits showed many internal changes in the paleoenvironment from deep sea to toe of slope, and slope inner platform-middle and outer platform.

Comparing the two different studied areas, where both Lower and Upper Cretaceous deposits were accumulated in quite different depositional conditions, it seems that Cretaceous sediments on Kastos Island were deposited in more stable and deeper conditions than those of Araxos peninsula, where several internal changes indicate shallower depositional conditions (Figure 7).

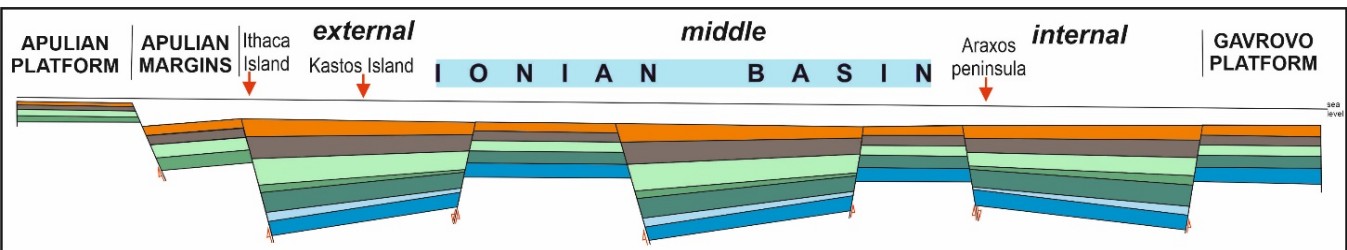

**Figure 7.** A schematic cross-section illustrating the Mesozoic depositional conditions of the Ionian basin, the geometry of troughs and highs, the impact of syn-sedimentary fault activity on depositional thickness, and the position of the studied areas, modified from [16].

According to geological and tectonic reconstructions [15], the western margins of the external Ionian sub-basin include both Ithaca Island (close to the western margins) and Kastos Island (central part of the sub-basin, according to this work). Therefore, based on their microfacies features, a comparison between Araxos peninsula (internal Ionian sub-basin) and Ithaca Island could be plausible as both areas represent the margins of a sub-basin very close to a platform from where breccia, microbreccia, or calciturbidites were produced. Additionally, as the above two areas are characterized by the presence of rudist and coral fragments, the above idea is supported.

Moreover, and regardless of microfacies analysis, the age determination of the studied samples showed great differences between previous geological mapping and the existing different rock formations.

In Kastos Island, several discrepancies were identified related to the studied outcrops. The outcrops of "Vigla shales" were found in restricted areas, thus they are not so thick and laterally extended as the published geological map shows (Figure 8).

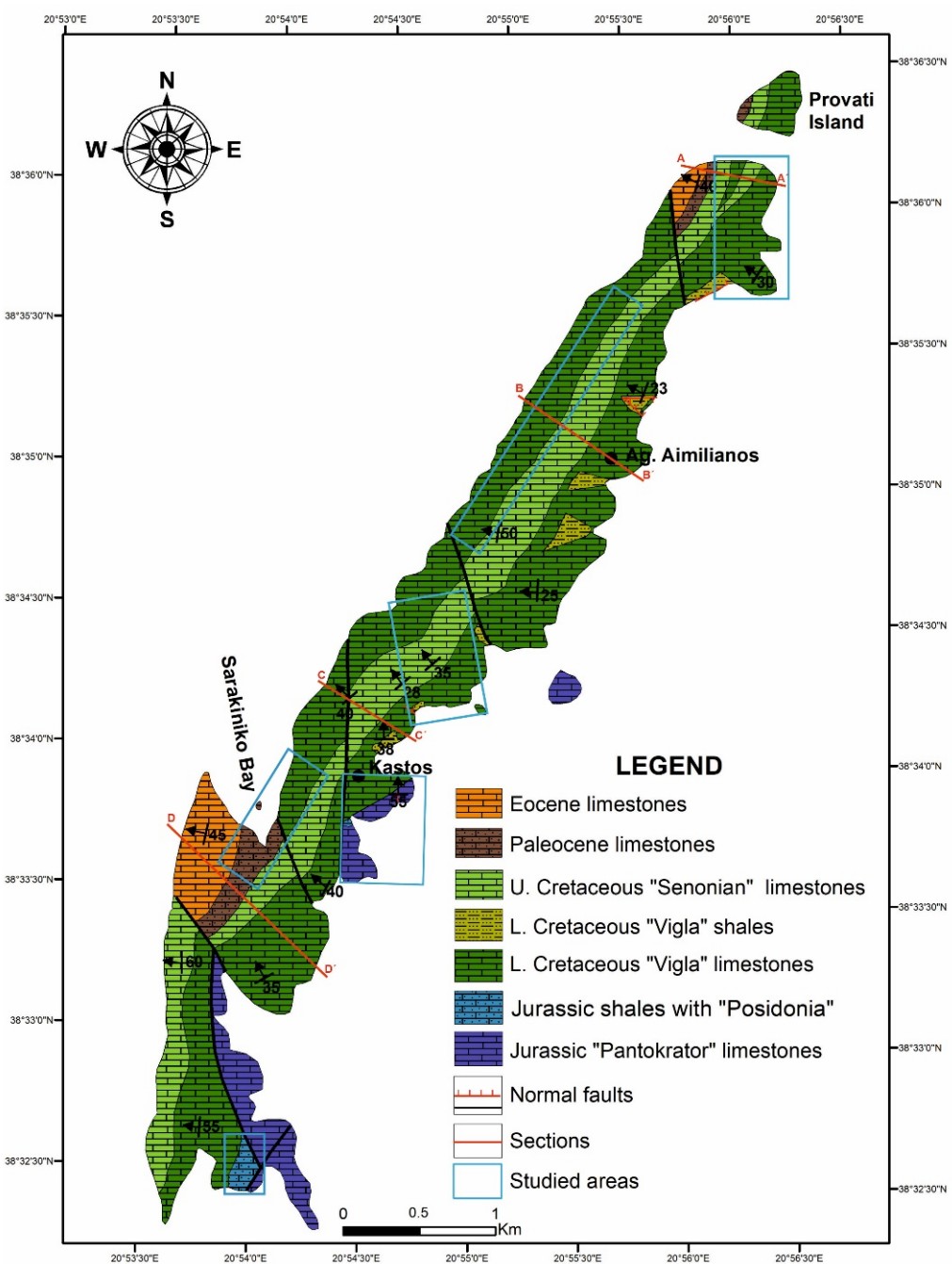

**Figure 8.** The final improved version of Kastos Island geological map. Note the absence of Paleocene and Eocene deposits in the center of the western side of the island. Cross-sections correspond to Figure 9, whereas blue boxes show the detailed studied areas with the selected samples.

The age determination from different outcrops east of Sarakiniko bay showed that Paleocene to Eocene deposits, as far as the northern part of Kastos Island is concerned, are absent. The fact that the western coastline of Kastos Island is characterized by "Vigla formation" introduces a new geological approach for the Kastos evolution (Figure 9).

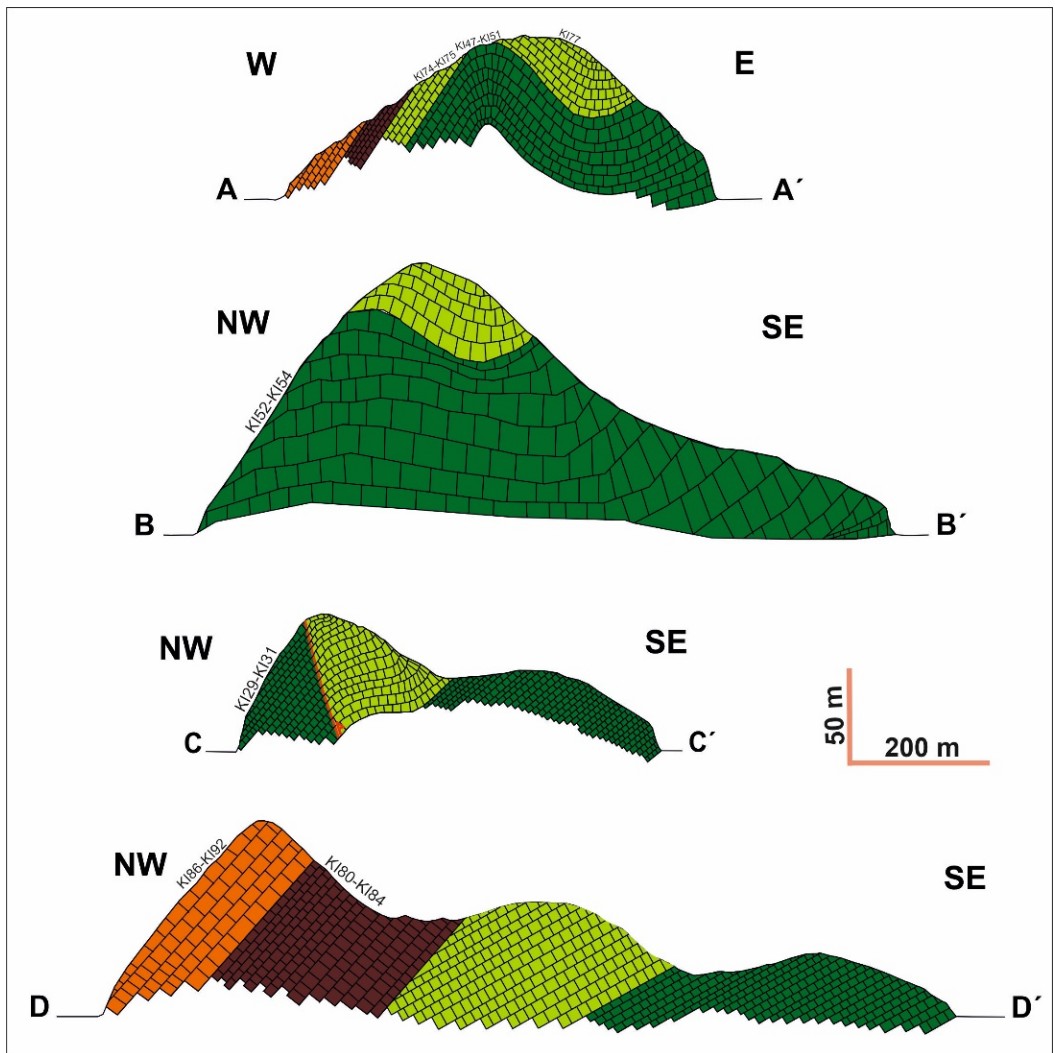

**Figure 9.** Four cross-sections show the present-day situation, from NE to SW on Kastos Island. For the location of cross-sections, see Figure 8.

In order to evaluate the new perception for Kastos Island evolution, four selected cross-sections were created presenting the new idea of Kastos evolution. It is critical to mention that the number of analyzed samples cannot support in total the proposed model and many new samples must be analyzed in order to find the exact boundaries between the different formations, to recognize the depositional changes either internal to each formation or between different formations.

In the new proposed geological map, two recent faults, as recognized and measured in the field, were added in order to help and establish the idea of a geological setting. With the above two new faults, an attempt was made in order to show the contact between different age formations. Additionally, Jurassic shales with "Posidonia" were found at least in one restricted area and this outcrop was also added to the new geological map.

Nevertheless, a question that has been raised from the new geological map and the existing additional faults is if all or some of the faults that appeared in the geological map are characterized by inversion tectonics. Most likely, they started their activity as normal faults, during Jurassic to Eocene, later acted as compressional faults, during Eocene to Miocene, and now transformed again to normal faults [15,43].

In Araxos peninsula, there are some minor internal changes to the Lower Cretaceous "Vigla formation," where "Vigla shales" were added to the geological map. The east-dipping bedding without any obvious unconformities from Lower Cretaceous to Eocene formations in relation with the fact that the oldest deposits are outcropping to the west

indicate the existence of a thrust fault west of the studied area (Figure 10), which acted after sedimentation and during the compressional regime, whereas the existing normal faults with west-dipping surfaces seem to be synsedimentary faults, and which acted during the syn-rift stage.

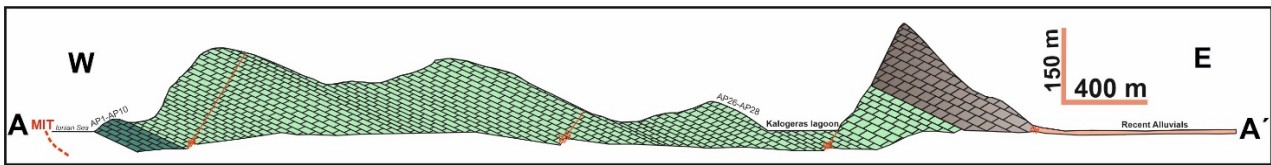

**Figure 10.** Cross-section showing the distribution of different formations in Araxos peninsula. For the location of cross-section see Figure 4, MIT: Middle Ionian Thrust (see Figure 1).

## 6. Conclusions

The goal of this study was the recognition of the different depositional conditions from one area to the other, both in the same age rocks and in different age rocks. Microfacies analysis results showed that on Kastos Island, the following depositional environments have been recognized: deep sea and deep shelf, toe of slope to slope, and open marine to restricted platform. In detail, the Lower Jurassic ("Pantokrator" limestones) deposits consist of SMF18-19 microfacies that were accumulated in an open marine-restricted environment (FZ7-8). The Lower Cretaceous deposits ("Vigla" limestones) are composed of standard microfacies SMF1-3, indicating deep sea to slope environments (FZ1-4), and in particular, the Valanginian ones by SMF3 and SMF1 microfacies are associated with deep sea environments (FZ1).

In Araxos peninsula, the Lower Cretaceous "Vigla" limestones consist of standard microfacies SMF3 and SMF2, indicating a deep-sea environment (FZ1-FZ2). The Upper Cretaceous microbreccia or breccia deposits consist of SMF5 microfacies which indicate a source of the exoclasts from a restricted or shallow shelf environment (FZ8) with the influence of meteoric water, whereas calciturbidites characterized bySMF4 microfacies and being interbedded by layers with radiolaria or planktonic foraminifera (SMF3), suggest slope or toe of slope depositional conditions (FZ3-4). Deep-sea conditions apply for the Paleocene calciturbidites containing planktonic foraminifera and radiolaria (SMF3), corresponding to a deep-sea environment (FZ1-2).

The above-described microfacies analysis and the recognition of facies zones introduce depositional changes in Araxos peninsula during the Cretaceous and several depositional changes on Kastos Island during the Paleocene/Eocene. Changes in depositional conditions could be related to different stages of tectonic activity in the Ionian basin from east to west. It seems that sedimentary conditions during the Cretaceous were clearly different between Kastos Island and Araxos peninsula, supporting the pre-existing results of our previously published works with different scientific approaches, such as the mineralogy of siliceous concretions or the tectonic influence (normal and transfer faults) on soft-sediment deformation during the sedimentation or on calciturbidites studies and their relation to the source areas [16,17].

Therefore, microfacies analysis introduced that the Cretaceous deposits were accumulated in quite different depositional conditions in the two studied areas. Kastos Island was characterized by more stable and deeper conditions and probably represents the deeper parts of the external Ionian sub-basin. On the other hand, Araxos peninsula was situated in the western margins of the external Ionian sub-basin, where shallower depositional conditions have been determined.

Finally, due to the results of this study, with fieldwork, microfacies analysis, and age determination the geological map of Kastos Island was reconstructed.

**Supplementary Materials:** The following are available online at https://www.mdpi.com/article/10.3390/geosciences11070288/s1, Figure S1: Coordinates of the selected samples from both areas.

**Author Contributions:** Conceptualization, N.B.; methodology, N.B.; G.I.; P.P.; data curation, P.P.; writing—original draft preparation, N.B.; A.Z.; writing—review and editing, G.I.; supervision, A.Z. All authors have read and agreed to the published version of the manuscript.

**Funding:** This research received no external funding.

**Data Availability Statement:** Data Availability Statements in section "MDPI Research Data Policies" at https://www.mdpi.com/ethics.

**Acknowledgments:** Nicolina Bourli was supported by "Andreas Mentzelopoulos Scholarships of the University of Patras" during her Ph.D. dissertation. We would like to thank Tadeusz Peryt and the two anonymous reviewers for their comments, which improved the final version.

**Conflicts of Interest:** The authors declare no conflict of interest.

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
