# Peer review of "Microfacies and Depositional Conditions of Jurassic to Eocene Carbonates: Implication on Ionian Basin Evolution"

_geosciences, doi:10.3390/geosciences11070288_

Round 1
Reviewer 1 Report
I was asked to newly review the work entitled “Microfacies and Depositional Conditions of Triassic to Eocene Carbonates: Implication on Ionian Basin Evolution” - authors: Nicolina Bourli, George Iliopoulos, Penelope Papadopoulou and Avraam Zelilidis. In fact, the paper was previously submitted to Geosciences as “Facies Analysis and Depositional Conditions of Mesozoic Carbonates: Implication on Ionian Basin Evolution” by the same authors, thus considered as a new submission by the Editor.
I accepted the Editor’s choice to ask me for a new revision, although I think it would be in the authors' interest to get comments from someone else as well.
To read critically a potential article and comment on it: the review procedure foresees this. The reviewers express their opinion, with respect to which one can certainly disagree.
I don't know any of the authors, and I doubt they know me. I do not deal with, nor have I dealt with in the past, the Ionian Basin, nor strictly carbonates. I have no interest in damaging or promoting the present research. I have read and commented on the text, I believe, with my head free from preconceptions and in the most objective possible way.
Therefore, I find the comment on my claimed conflict (of interest or of what kind?) rather ungenerous and at least unkind. I understand it doesn't please anyone to receive criticism, but as the authors, I would have been more cautious in some statements. I want to assume that they have misunderstood my identity, and it seems to me that there is some “scientific” dispute going on in the Greek scenery. Frankly, I don't want to fight any Peloponnesian war, nor do I have any enemies to stop at Thermopylae. My comments, here and in the manuscript, remain absolutely personal opinions, on which the authors and the Editor can agree or not. This applies to old comments as well as new ones. Anyway, to assert “we followed their points that are not in different direction from our thoughts” sounds poorly scientific if it is not argued.
Below, I will integrate reply to the authors (old version) and comments to the new manuscript. Authors can find proposed/required corrections and minor comments in attachment. By the way, both general overview (here) and line-by-line review (attached file) contribute to review procedure.
- As in the previous version, the paper deals on microfacies analysis and biostratigraphy of Kastos Island and Araxos Peninsula (Ionian basin), with the aim to reconstruct the Mesozoic-Cenozoic paleo-depositional environments and their evolution, in relation with tectono-sedimentary phases. I am still of the opinion that this research provides new data and interesting insights about the paleoenvironmental evolution of the Ionian Basin, as well the manuscript needs some improvements. It is not my intention to destroy anything, but the data and interpretations should be presented adequately. I recognize that, if compared to the previous version, the new manuscript is stylistically closer to the others previously published by the same authors, still on issues related to the Ionian Basin. Parts more related to a Thesis or a report have been eliminated, favouring the state of the art and the discussion. Nonetheless, I am still of the opinion that some of the proposed conclusions are poorly explained in light of the data presented, and still I feel that some aspects are taken for granted or otherwise acquired. I again suggest to the authors to consider the conclusions reached in their recent works, and to propose or reconsider them in this context. It is true that these works deal with other themes (calciturbidites, mineralogy of siliceous concretions and soft-sediment deformations), but they are still a piece of the mosaic that the authors have in mind and are recomposing. The reference to these previously published papers was simply linked to this: allowing the reader to understand what your fixed points are and what this work adds to the reconstruction of the general picture, what are the limits and in which direction the line of research is going. If authors “believe that the present research improve our knowledge and the previous results as it offers new updated results”, this should be better highlighted.
- I believe authors when they say “(English) specialists have updated the manuscript”, yet check minor points (mollusk vs. mollusc, palaeo vs. paleo, etc.).
- Please check the repeated spaces, and the use of italics for genera/species (just typos, I think).
- Abstract, Introduction, Results paragraph have been largely modified according to my previous comments, although I cannot verify if and how the requests of the other reviewer have been accepted.
- I think I understand the authors' motivations for focusing on two well-defined areas; nonetheless, I still think something could be added, although limited to some specific stratigraphic units, about the following points:
1) association between recognized microfacies/micropaleontological assemblages, codified facies zones and paleoenvironments.
2) sedimentological/stratigraphic columns, vertical/lateral facies variations, mesoscale information (geometry of beds, sedimentary structures, macrofossils and their taphonomy), etc. (field work results).
3) implications for tectono-sedimentary phases and paleogeographic restoration.
I believe that adding, or deepening, these aspects would further improve the text, broadening its scope; anyway, it is authors’ choice.
- I still found both the methodological choice applied, and the conclusions proposed in the light of the provided data a little weak. On this last aspect, it seems to me that the authors also agree on the need to collect further data. I believe that the topic presented is certainly interesting, as well as the manuscript still could to be improved.
Anyway, with some perplexity on manuscript present form, I have no substantial reasons to oppose publication.
Thus, I proposed the Editor to accept the manuscript with minor revisions.
Best Regards
The reviewer

Author Response
We accepted all grammar and spelling suggestions as these suggested and marked in their submitted PDF file.
We improved most chapters (introduction, material and methods, results, conclusions), as suggested, giving more details or omitting some parts of the text (see resubmission with track-changes).
In detail:
1) association between recognized microfacies/micropaleontological assemblages, codified facies zones and paleoenvironments.
2) sedimentological/stratigraphic columns, vertical/lateral facies variations, mesoscale information (geometry of beds, sedimentary structures, macrofossils and their taphonomy), etc. (field work results).
We agree with the point 2 of reviewer 1 that it is necessary to have “sedimentological/stratigraphic columns, vertical/lateral facies variations, mesoscale information (geometry of beds, sedimentary structures, macrofossils and their taphonomy), etc. (field work results)”, but we can build small columns when we have the possibility to do it. In our cases, there are good exposures only in the eastern side of Kastos Island where mostly or only the Vigla formations is observed. Therefore, when we studied only Cretaceous deposits (see Bourli et al., 2020) we presented details about this formation. In this study, we believe that it will not be correct to present only details for one formation and only for lower Cretaceous, as the stratigraphic column of Kastos Island produced synthetically from many small outcrops.
3) implications for tectono-sedimentary phases and paleogeographic restoration.
We agree that this is the key to give some information about tectono-sedimentary and paleogeographic restoration with your results. Taking into account that the Ionian Basin is too long, wide, and influenced both from many normal and from transfer faults activity, during the syn-rift stage, we believe that any attempt would be very dangerous, from our point of view, and for this reason, we avoid it.
I believe that adding, or deepening, these aspects would further improve the text, broadening its scope; anyway, it is authors’ choice.
- I still found both the methodological choice applied,
and the conclusions proposed in the light of the provided data a little weak. On this last aspect, it seems to me that the authors also agree on the need to collect further data. I believe that the topic presented is certainly interesting, as well as the manuscript still could to be improved.
Anyway, with some perplexity on manuscript present form, I have no substantial reasons to oppose publication.
See 2.2 Studied Areas new lines 178-183,
- Materials and Methods new lines 233-238.
- Results new lines 248-250, 383-391, omitted lines 251-260, 269-312, 331-351, 392-451
- Discussion new lines 512-513, 551-553
- Conclusion new lines 580-585
The suggestions to include some of thin-sections information from different studied regions in new tables, including or separating them into studied regions, as we believe that in that cases we will confuse readers about the stratigraphy in total, and in the internal changes either in ages or in depositional conditions, isn’t followed.

Reviewer 2 Report
The paper presents a study of the Triassic to Eocene carbonate microfacies of Kastos Island and the Araxos Peninsula in Greece. Field observations including stratigraphic survey and geological mapping are well conducted. Also the analysis of microfacies and microfaunas on samples are convincing.
However, the scientific objective of the paper is not clear. It is not clear what scientific interest there is in comparing two sections more than 60 km apart and located in different tectonic units. Nothing is said in this sense in the introduction and the conclusion simply leads to a definition of the depositional environments in each area and an improvement of the Kastos geological map. The data and analyses are interesting; what is needed now is to define a clear scientific question that can be answered with these data.
Some miscellaneous remarks on the manuscript. 1- the Introduction: the part between line 29 and line 63 is useless. These are well-known considerations that can be found in all carbonate sedimentology textbooks. Moreover, Folk's classification should no longer be used. Sparite is a cement (not a matrix), which cannot give any indication of the depositional environment. 2- Geological setting: the description of the formations is too detailed and complex. Simplify it by putting the essential information in a figure. On the other hand, be more precise about the general geodynamic evolution. 2.2 studied area: Specify the link between the two studied areas. Why these two areas? 3 Material and methods: Add a more precise field log showing the banks, lithology and textures of the rocks and the position of the samples. Otherwise, it is not clear how taking a sample every 10m (Kastos) or every 15m (Araxos) is scientifically justified. Was this sampling step chosen at random or was it based on the facies changes observed in the field? 4 Results: Difficult to read. The results presented in the text can be integrated into the tables. Only the main results should be presented here, i.e. depositional environments, ages and the geological map. 5- Discussion: The outline of the discussion can only be determined after the objective of the paper has been defined
Author Response
We improved most chapters (introduction, material and methods, results, conclusions), as suggested, giving more details or omitting some parts of the text (see resubmission with track-changes).
Some miscellaneous remarks on the manuscript. 1- the Introduction: the part between line 29 and line 63 is useless. These are well-known considerations that can be found in all carbonate sedimentology textbooks. Moreover, Folk's classification should no longer be used. Sparite is a cement (not a matrix), which cannot give any indication of the depositional environment. 2- Geological setting: the description of the formations is too detailed and complex. Simplify it by putting the essential information in a figure.
We disagree with reviewer 2 for his point mentioning that the text from lines 29-63 is useless, because first we have to show how we used the literature but mostly to show researchers what they have to read or how to work if they would like to work with equivalent subjects, as are not all readers familiar to the subject. In the same sense and according to the most detailed presentation, this of Flügel 2010, where it is stated that Folk’s classification is still in use and new classifications were based and on his work. Finally, yes, sparite is cement and not matrix but can give valuable information for depositional conditions such as the energy of the environment. See what Flügel 2010 wrote: The basic philosophy of Folk’s classification is that carbonate rocks are similar to siliciclastic rocks in their mode of deposition, because their textures are both controlled largely by the water energy in the depositional area. In calm water with only sluggish currents carbonate mud (represented as micrite) with or without grains is predominantly deposited. By contrast, vigorous water energy hinders deposition of fine-grained material, thus favoring the sedimentation of winnowed sands with large amounts of pore space that is later filled with sparry calcite (sparite). The most important environmental break is between limestones with a lime-mud matrix and those with calcite cement, because this should reflect the point where water energy becomes turbulent enough to wash out (winnow) the lime mud, keep it in suspension and carry it into lower energy zones.
On the other hand, be more precise about the general geodynamic evolution. 2.2 studied area: Specify the link between the two studied areas. Why these two areas? 3 Material and methods: Add a more precise field log showing the banks, lithology and textures of the rocks and the position of the samples. Otherwise, it is not clear how taking a sample every 10m (Kastos) or every 15m (Araxos) is scientifically justified. Was this sampling step chosen at random or was it based on the facies changes observed in the field? 4 Results: Difficult to read. The results presented in the text can be integrated into the tables. Only the main results should be presented here, i.e. depositional environments, ages and the geological map. 5- Discussion: The outline of the discussion can only be determined after the objective of the paper has been defined
See 2.2 Studied Areas new lines 178-183,
- Materials and Methods new lines 233-238.
- Results new lines 248-250, 383-391, omitted lines 251-260, 269-312, 331-351, 392-451
- Discussion new lines 512-513, 551-553
- Conclusion new lines 580-5857.
The suggestions to include some of thin-sections information from different studied regions in new tables, including or separating them into studied regions, as we believe that in that cases we will confuse readers about the stratigraphy in total, and in the internal changes either in ages or in depositional conditions, isn’t followed.

Reviewer 3 Report
I attach the file with my comments.

Author Response
A. Most of his/her comments related with the use of ages descriptions. Although we believe that we followed the correct way and in order to avoid new disagrees from the reviewer we followed all his/her suggestions either in the text or in the figures.
B. Concerning the comments in introduction chapter:
We disagree with reviewer’s comments because first we have to show how we used the literature but mostly to show researchers what they have to read or how to work if they would like to work with equivalent subjects, as are not all readers familiar to the subject. In the same sense and according to the most detailed presentation, this of Flügel 2010, where it is stated that Folk’s classification is still in use and new classifications were based and on his work. Finally, yes, sparite is cement and not matrix but can give valuable information for depositional conditions such as the energy of the environment. See what Flügel 2010 wrote: The basic philosophy of Folk’s classification is that carbonate rocks are similar to siliciclastic rocks in their mode of deposition, because their textures are both controlled largely by the water energy in the depositional area. In calm water with only sluggish currents carbonate mud (represented as micrite) with or without grains is predominantly deposited. By contrast, vigorous water energy hinders deposition of fine-grained material, thus favoring the sedimentation of winnowed sands with large amounts of pore space that is later filled with sparry calcite (sparite). The most important environmental break is between limestones with a lime-mud matrix and those with calcite cement, because this should reflect the point where water energy becomes turbulent enough to wash out (winnow) the lime mud, keep it in suspension and carry it into lower energy zones.
- Detailed changes on the text (see resubmission with track-changes).
See 2.2 Studied Areas new lines 178-183,
- Materials and Methods new lines 233-238.
- Results new lines 248-250, 383-391, omitted lines 251-260, 269-312, 331-351, 392-451
- Discussion new lines 512-513, 551-553
- Conclusion new lines 580-5857.
The suggestions to include some of thin-sections information from different studied regions in new tables, including or separating them into studied regions, as we believe that in that cases we will confuse readers about the stratigraphy in total, and in the internal changes either in ages or in depositional conditions, isn’t followed.

Round 2
Reviewer 2 Report
I see no change in the new manuscript from the old one. The letter indicates changes with line numbers that do not correspond in the new manuscript. Revision required
Author Response
I would like to inform you that in our previous pdf file there were two files, one with truck changes and a second one without the trucks. Reading the file with the trucks you can see how many changes were made mostly according to the other two reviewers. I believe that when a reviewer suggests moderate revisions have also to show in a pdf the changes that would like to do. It is unfair to suggest moderate revision without stating what you want to change.
Therefore, the new pdf is according to the other reviewers suggestions.

Reviewer 3 Report
Many editorial corrections are needed as is indicated in the annotated file.
Tadeusz Peryt

Author Response
We tried to follow your suggestions, from your PDF file, in order to improve manuscript quality.
Moreover, I would like to inform you that most of your suggestions according to presentation of fossils (italics) there were in correct form (see our previous pdf file) and probably as the journal changed our Ms from word to pdf did not accept the character of italics.
In any case, thank you very much, for your detail reviewing.

Round 3
Reviewer 2 Report
I am very sorry but the authors — from what I can see in the new manuscript and the authors' cover letter — do not follow at all my recommendations. The majors ones were (1) the lack of any scientific question in the introduction and answer in the conclusion, (2) the misleading use of the Folk classification and (3) most, if not all, the miscellaneous comments. Despite the fact that the line numbers mentioned in the cover letter do not correspond to anything in the new text version.
Author Response
Concerning the reviewers comments with his/her three major comments our reaction is as follow:
Comment 1: The lack of any scientific question in the introduction and answer in the conclusion,
Although we disagree with this point, we tried to present clearest the target-question in the introduction and after this to present also clearest the goal-answer in the conclusion.
See new lines 62-67, 419-420 and 445-449 in the text without trucks and with yellow background.
Comment 2: The misleading use of the Folk classification and (3) most, if not all, the miscellaneous comments.
We improved our ms, following the instructions of the reviewer, showing the use of Folk classification.
Improved lines 252-266 and 298-307 in the text without trucks and with yellow background
Comment 3: Despite the fact that the line numbers mentioned in the cover letter do not correspond to anything in the new text version.
We disagree with the reviewer point that our changes in the previous text are not clearly presented.
I would like to mention that in our previous ms and according the other two reviewers’ suggestions the text modified considerably, and the above changes have been accepted from the two reviewers.
In detail,
- In the new, previously submitted ms, the text that was added was:
- After our previous studies [15-17]that showed different depositional conditions either due to their different tectonic activity or due to the different influence of the neighboring existing platforms (Apulian to the west and Gavrovo to the east), the same two areas were selected for the detailed study of this work. The above differences referred to different mineralogy of siliceous concretions, different kind of soft-sediment deformations and different influence of tectonic activity (normal or transfer faults) in Ionian basin depositional conditions.
- As it was very strange Kastos Island to present generally no deformation during compressional regime and Araxos peninsula to be influenced from a thrust fault, during compressional regime, situated west of the studied area [15-17] first sampling was focuses on the age determination and facies analysis in specific areas, where previous published works highlighted. After the first results, additional sampling focuses on the transitional zones between different age formations in order to highlight the depositional conditions changes.
- (this text changed position). The main textural and compositional characteristics of the studied total thin sections, as well as the sedimentary features of the distinguished microfacies are presented in Table 1, corresponding to different depositional environments or standard facies zones [5].
- More specifically, the Lower Cretaceous deposits (Vigla limestones) were classified as mudstone/wackestone limestones with planktonic foraminifera and radiolaria (standard microfacies SMF1-3), indicating deep sea to slope environment (FZ1-4). In the Lower Cretaceous (Valanginian) the observed mudstone-wackestone limestones, consist of biomicrites with skeletal grains and spiculites (SMF3, SMF1) and are associated to as a deep-sea environment (FZ1).
Calciturbidites of Late Cretaceous were classified as packstones with microbreccia and planktonic foraminifera (SMF4) in a slope environment (FZ4).
The Paleocene deposits consist of grain supported wackestone-grainestone biomicrites with skeletal grains.
- (Figure 8), acted after sedimentation and during the compressional regime, whereas the existing normal faults with west dipping surfaces seems to be synsedimentary faults, acted during syn-rift stage.
- supporting the pre-existing results of our published works with different scientific approach, like the mineralogy of siliceous concretions or the tectonic influence (normal and transfer faults) on soft-sediment deformation during the sedimentation or on calciturbidites studies and their relation to the source areas.
Due to the results of this study, with fieldwork, microfacies analysis and age determination the geological map of Kastos Island was reconstructed.
In the new, previously submitted, ms the text that was omitted is:
In detail, the following microfacies were determined and their supported facies zones that were recognized (Table 1), are:
North Kastos Island:
- Early Cretaceous (Berriasian-Valanginian): Mudstone/Wackestone, skeletal grains, fossiliferous biomicrite (SMF3), deep sea (FZ1).
- Early Cretaceous (Aptian-Albian): Mudstone with calcite veins and stylolites, fossiliferous biomicrite (SMF3), deep sea (FZ1)
- Early-Middle Paleocene (Danian-Selandian): Packstone, microbrecciated, few skeletan grains, microsparite (SMF4), slope (FZ4).
West Kastos Island:
- Early Cretaceous (Late Aptian): Wackestone/Mudstone, locally lamination (SMF3), deep sea (FZ1).
- Late Cretaceous (Late Cenomanian): Mudstone (SMF3), deep sea (FZ1).
Central Kastos Island:
- Early Cretaceous (Valanginian): Mudstone/wackestone, skeletan grains, biomicrite, stylolites (SMF3), deep sea (FZ1).
- Early Cretaceous (Aptian): A. Mudstone to wackestone, microbreccia, B. Packstone/grainstone, stylolites (SMF2), deep sea (FZ1).
- Early Cretaceous (Albian): Mudstone, skeletal grains, biomicrite (SMF3), deep sea (FZ1).
- Late Cretaceous (Turonian): Grainstone, microbrecciated (SMF5), slope (FZ4).
- Late Cretaceous (Maastrichtian): Wackestone/Packstone, microbrecciated, skeletan grains (SMF4), toe of slope (FZ3).
Sarakiniko bay:
- Early Cretaceous (Aptian): Packstone, microbioclastic, microcrystalline (SMF2), deep sea-deep shelf (FZ1-2).
- Early Cretaceous (Albian): Wackstone/Packstone, microbioclastic, microcrystalline with lamination (SMF3), deep sea (FZ1).
- Middle-Late Paleocene (Selandian-Thanetian): Wackestone with sparse biomicrite skeletan grains, foraminifera, calcite veins (SMF3), deep sea (FZ1).
- Early Eocene (Ypresian): Wackestone/Packstone, biomicrite, skeletal grains, calcite vein (SMF3), toe of slope (FZ3).
- Early Eocene (Lutetian): Two facies: a. Grain supported, packstone, biosparite, peloids, calcite vein, peloids, plaktonic foraminifera, microcrystalline calcite, b. Mudstone/Wackestone, biοmicrites, skeletan grains (SMF3), toe of slope (FZ3).
- Middle-Late Eocene (Bartonian-Priabonian): Mudstone/wackestone (mud to grain supported), sparse micrites, skeletan grains (SMF3), deep sea (FZ1).
Kastos village
- Early Jurassic: Wackestone/grainstone and sometimes boundstone (SMF18), open marine-restricted (FZ7-8).
- Early Cretaceous: Mudstone (SMF3), deep sea (FZ1).
South Kastos Island:
- Early Jurassic: a. Grainstone with peloids (pelsparite) (SMF18) open marine-restricted, b. Mudstone with lamination, stylolites. Facies A probably an exoclast from pantokrator formation limestone (SMF2), deep sea (FZ1).
- Late Jurassic: Mudstone (SMF3), deep sea (FZ1).
- Early Cretaceous: Mudstone, stylolites (SMF3), deep sea (FZ1).
The main textural and compositional characteristics of the studied total thin sections, as well as the sedimentary features of the distinguished microfacies are presented in Table 1, corresponding to different depositional environments or standard facies zones [5].
- Early Jurassic: Thaumatoporella , Siphovalvulina sp., Miliolidae, Triloculina sp.
- Late Jurassic: Radiolaria, rare Bivalve filaments, Calpionellidae, Calpionellites , Tintinnopsella sp.
- Early Cretaceous (Valanginian): Calpionellidae, Lorenziella cf. hungarica, Tintinnopsella carpathica, Clavihedbergella eocretacea, Praehedbergella sigali
- Early Cretaceous (Aptian): Blefuscuiana gorbachicae, Blefuscuiana occulta, Blefuscuiana praetrocoidea, Globigerinelloides cf. ferreolensis, Globigerinelloides gottisi, Globigerinelloides barri, Globigerinelloides blowi
- Early Cretaceous (Albian): Radiolaria, Ticinella cf. praeticinencis, Hedbergella planispira
- Late Cretaceous (Cenomanian): Radiolaria, Clavihedbrgella simplex, Hedbergella planispira, Rotalipora cushmani, Praeglobotruncana delrioensis, Thalmanninella appenninica, Thalmanninella, greenhornensis, Thalmanninella globotruncanoides, Whiteinella archaeocretacea
- Late Cretaceous (Turonian): Helvetotruncana helvetica
- Late Cretaceous (Maastrichtian): Radiolaria, Rugoglobigerina sp., Globotruncana arca, Globotruncana neotricarinata, Globotruncana orientalis, Globotruncanita cf conica
- Paleocene: Morozovella angulata, Planorotalites pseudomenardi, Morozovella aequa Morozovella conicotruncana Planorotalites chapmani Subbotina inaequispira
- Eocene: Radiolaria, Bivalve filaments, Globigerinotheka , Turborotalia sp. Turborotalia cerroazulensis, Catapsydrax cf. dissimilis, Pseudohastigerina micra Globigerina sp., Catapsydrax sp., Acarinina sp.
Gianiskari beach
- Early Cretaceous (Tithonian-Valanginian): Mudstone-wackestone (SMF3), deep sea (FZ1).
- Early Cretaceous (Valanginian): Wackestone (SMF3), deep sea (FZ1) and mudstone with biomicrites (SMF3) and deep sea (FZ1).
- Early Cretaceous (Albian): Wackestone with bioclasts (SMF3), deep sea (FZ1).
- Middle Cretaceous: Mudstone (SMF1), deep sea (FZ1).
- Late Cretaceous (Senonian): Mudstone/wackestone (SMF3), deep sea (FZ1).
On the road
- Late Cretaceous (Turonian-Santonian): Packstone/wackestone with small-sized bioclasts (SMF4), slope (FZ4)
- Late Cretaceous (Campanian-Maastrichtian): Wackestone with scattered benthic foraminifera, echinoderm fragments and pelloids. Fenestral cavities maybe due to meteoric diagenesis (SMF19), restricted platform (FZ8) (most likely an exoclast). Allochthonous bioclastic rudstone/ floatstone breccia, geopetal fractures (SMF5), slope (FZ4).
- Late Cretaceous (Maastrichtian): Packstone with bioclasts (SMF2), deep sea/deep shelf (FZ1/FZ2).
Quarry
- Late Cretaceous (Santonian-Maastrichtian): Packstone with bioclasts (SMF4), slope (FZ4), wackestone/packstone with material derived from reef organisms (SMF4), slope (FZ4).
- Late Cretaceous (lower Campanian): Microbrecciated limestone with abundant bioclastic fragments and micritic clasts, ooids fragments, geopetal fabrics (SMF5), slope (FZ4).
- Late Cretaceous (Campanian): Packstone/wackestone with endoclast (SMF4), toe of slope (FZ3).
- Late Cretaceous (Campanian-Maastrichtian): Allochthonous bio-lithoclastic packstone/rudstone, oosparite endoclast, micritic endoclast (SMF5), slope (FZ4). Also, in places, grainstone/rudstone (higher energy). Floatstone/Rudstone with bioclasts of rudists and mollusk fragments, intraclasts with benthic foraminifera. Environment with high energy (SMF5), slope (FZ4). Locally, allochthonous bio-lithoclastic packstone. Coarse crystalline mosaic of calcite with micritic clasts and fenestral caves. Also, the typical dog-teeth cement has developed around some breccia and suggests meteoric late diagenesis (SMF5), slope (FZ4).
- Late Cretaceous (Maastrichtian): Floatstone with bioclasts of rudists, benthic foraminifera from a nearby shoal (SMF5), slope (FZ4). Wackestone and calcite veins filled with coarse calcite grains. Calciturbidite (calcilutite), Wackestone-Packstone with small-sized bioclasts (rudists, echinoderms) (SMF4), slope (FZ4) and pelagic mudstone with Globotruncana (SMF3), deep sea (FZ2).
- Late Cretaceous (Senonian): Micro brecciated limestone with nodules just over decollement surface (SMF4), slope (FZ4). Wackestone with a packstone lamination with planktonic foraminifera (SMF3), toe of slope (FZ3) environment.
- Paleocene (Selandian): Pelagic wackestone with planktonic foraminifera (SMF3), deep sea (FZ2)
- Paleocene (Selandian-Thanetian): Pelagic wackestone with planktonic foraminifera. Most of the tests of the planktonic foraminifera are oriented parallel to the bedding. Calcite veins. (SMF3), deep sea/deep shelf (FZ1/FZ2).
The biostratigraphic analysis, on the selected thin sections, revealed the following assemblages (Table 2):
- Early Cretaceous: Calpionella , Radiolaria, Ticinella sp.
- Late Cretaceous (Late Santonian): Radiolaria, Rugoglobigerina, Marginotruncana pseudolinneiana, Planomalina sp., Hedbergella sp.
- Late Cretaceous (Santonian to Maastrichtian): Quinqueloculina , Spiroloculina sp., Pseudolituonella sp., Cuneolina sp., Textulariidae, Miliolidae, Bolivinopsis sp., Globotruncana cf. arca, Mollusc fragments.
- Late Cretaceous (late Maastrichtian): Radiolaria, Rudist fragments, other Mollusc fragments, Dasycladacea, Miliolidae, Cuneolina , Orbitoides cf. media, Orbitoides apiculata, Globotruncanita stuarti, Rugoglobigerina rugosa, Globotruncana cf. bulloides, Globotruncana arca, Abathomphalus mayaroensis, Contusotruncana sp.
- Paleocene (Selandian-Thanetian): Radiolaria, Parasubbotina pseudobulloides, Acarinina, Subbotina sp., Igorina pussila, Chiloguemberlita sp., Morozovella aequa.
In the new ms without truck changes we marked with yellow shadow the new updated text as mentioned about comments 1, 2.

This manuscript is a resubmission of an earlier submission. The following is a list of the peer review reports and author responses from that submission.
Round 1
Reviewer 1 Report
Bourli et al. manuscript titled “Facies Analysis and Depositional Conditions of Mesozoic Carbonates: Implication on Ionian Basin Evolution” stated objective is to define the depositional environments of the carbonate Mesozoic sequence of the Ionian Basin. For that objective, the authors elected to use the “standard microfacies” model proposed by Flügel (2004). This model is handy, but its application here is fundamentally flawed. The concept of the “standard model” first proposed by Wilson (1975) is a framework into it the specific facies observed could be fitted. The specific of the facies are location and time depended. Given that we further understanding disintegrated the “facies belts” into “facies mosaic” shifted the clustering of facies into “facies zones” clustering different facies together. All of this, and the critic of the basic concept are outlined in chapter 14 of Flügel (2004). There the standard microfacies types are defined as “virtual categories that summarize microfacies with identical criteria” and noted the SMF he outlines were “were defined for a model describing the sedimentation on a rimmed carbonate shelf and warm-water platform-reef environments in tropical latitudes.” (as such, this setting must first be established). These SMF are useful generic categories to fit the observed facies, but to do so, the observed facies must first be properly described. One cannot just skip to assigning SMF without describing the facies, delineating the possibilities and assigning the sample to these categories based on a discussion. This is irrespective of the fact that this model does not properly account for temporal elements which can shift specific groups to deeper and shallower waters (see discussion in various papers by Flügel, Pomar, Wood and many others).
Moreover, what the authors present here is less of a proper facies work and more of an addendum to a geological map where a few type sections are described given that much of the discussions has to do with lithostratigraphy rather than inferring depositional environments. If this work would have been recontextualized in that framework it’ll have more merit, although its interest to a non-local reader will be limited. Nevertheless, we do need more detailed and well described geological maps and I strongly encourage this work to be resubmitted with a full GIS database.
Figures and tables:
Figure 1 - needs an inset map, at least at the scale of the Eastern Mediterranean, showing the location of the study area and coordinates should be in lat/long degrees. At present, there is no way of knowing where the study area is from this map.
Figure 2 – add a legend to this figure, it is illegible in its current forms. Moreover, I strongly recommend it would be redrawn as the swaths are identical across multiple units, making it indiscernible to a colour-blind person (or black and white print).
Figures 3, 4 & 6 – add lat/long coordinates
Plates 1 & 2 – noted where and from what age is each sample. See about improving the lighting in these photomicrographs, note if the images are CX or PPL.
Tables 1 & 2 – this table cannot be read the font is too small and too condense to be legible, additionally, lat/long coordinates are needs for all samples. I would strongly recommend splitting the fossils into “age determining fossils” and “other fossils present”. The type of fossil in the former, as well as age and reference (e.g. Globigerina falconensis [F], <18Ma, Kennett & Srinivasan 1983) are a must. If needed move this table to the supplement as a spreadsheet.
Reviewer 2 Report
Dear Authors,
I was asked to review the work entitled “Facies Analysis and Depositional Conditions of Mesozoic Carbonates: Implication on Ionian Basin Evolution” - authors: Nicolina Bourli, George Iliopoulos, Penelope Papadopoulou and Avraam Zelilidis.
The paper focuses on microfacies analysis and biostratigraphy of two selected areas at the margins of the Ionian basin, with the aim to reconstruct the paleo-depositional environments and their evolution, particularly their relation with pre-rift and syn-rift tectono-sedimentary phases, across the Cretaceous and the Eocene.
In my opinion, this research undoubtedly contains interesting insights about the paleoenvironmental evolution of the Ionian Basin, and in particular regarding the relationship between tectonic and sedimentation during the Late Cretaceous-Paleocene. Nonetheless, some issue arises.
The same authors recently published various articles on the same area, focusing on themes contiguous or partially overlapping to the subject of the present research. Some of these works are also appropriately mentioned in the references. This in itself would not be prejudicial to article’s publication, provided that it is well defined how it adds new elements. Unfortunately, this is difficult to evaluate at the moment, as some crucial elements are not made available to the reader. Some of the proposed conclusions, indeed, are little or nothing justifiable in the light of the few data presented, and there is a feeling that some aspects are taken for granted or otherwise acquired (i.e.: already published?).
Thus, the manuscript shows these and other weaknesses, which do not allow the publication in its present form. I believe that the topic presented may certainly be interesting, but the manuscript still needs much work from the authors. Thus, I proposed the Editor to reject it.
Authors can find my line-by-line comments in the text (see the attached file). Here, some general comments and suggestions:
- Not being a native English speaker, I feel not enough qualified to correct the linguistic/stylistic aspects: anyway, I suggest the authors to check the choice of terms (which do not always seem appropriate), as well as the length and articulation of some sentences, which appear difficult to read.
- The article seems to arise from a Thesis work, or from a geological survey, or both. Usually I really appreciate the spirit of these works, which are born directly on the field. However, the data collected and their interpretation, with the following modelling, should be presented in a different way. This sounds a bit strange, comparing it to the aforementioned recent publications by the same authors, and some choices in the organization and presentation of data can be hardly explained. For example, I find it unnecessary in this context to focus on the importance of acquired classification schemes for carbonate rocks, and coded descriptive and/or analytical protocols, unless the purpose of the article is really a critical review of such schemes. Rather, it should be clarified why reference is made to some authors and not to other, in the light of the interpretation given to the collected data. Otherwise, it seems like an academic exercise that is irrelevant to a scientific article. I mean: we all use the Folk and/or Dunham classifications, just as we apply procedures like the one suggested by Tucker, there is no need to describe them or stress their importance. Instead, I would go and compare from time to time the microfacies described in this work both with the codified ones (authors refer to Wilson, 1975, and to Flügel, 2004, 2010: feel their schemes conclusive for the purpose?), and with other case studies (why do not consider other authors?), pointing out the paleoenvironmental interpretation that are the most relevant to this research.
- The Abstract merely lists recognized paleoenvironments, suggesting that they may shed new light on the geological reconstruction of the Ionian Basin, but not how and why.
- The Introduction paragraph lacks of a real “state of the art”: it is not possible for the reader to determine where this work falls within the general framework of knowledge on the subject. If the work aims, starting from the analysis of microfacies, at the paleoenvironmental reconstruction and at an evolutionary geological model of the area studied within the Ionian basin, these two aspects must be introduced here. At this point, the introduction of the case study is almost automatic: within which specific geological problem does this research fit, how this problem has been previously addressed, what new contribution is provided here, what implications it may have for the pure and applied research, even in a wider context.
- The main part of the paper (“Data analysis and their results”) lists the description of recognized microfacies and micropaleontological assemblages, distributing them among various codified facies zones and thus attributing them to different paleoenvironments. This procedure (not the only possible one, however) is only briefly described, without discussing the advantages and limits of this choice. In my opinion, this should be added in the “Material and Methods” paragraph. Even if (supplementary material) the studied samples are geo-referenced (latitude, longitude), they should be placed inside sedimentological/stratigraphic columns, in order to allow the reader to value the vertical and lateral facies variations. In the two study areas, about 90 and 40 samples were analyzed through a total thickness of up to 800 m and 550 m, respectively: it is not possible to say whether this is enough for a paleoenvironmental and evolutionary reconstruction, without the aforementioned stratigraphic positioning. Moreover, without local columns the mesoscale information (geometry of beds, sedimentary structures, macrofossils and their taphonomy, etc.) totally lacks. A brief part of field work results is needed, I think. If such description was already reported (i.e.: it is part of a Thesis or of a previous paper) add citations. Otherwise, explain why you focused on microfacies. Above all, a graphical, presentation of the results would benefit the work. Finally, rather than list “Microfacies and their supported Facies Zones”, you should explain why some microfacies and microfossil assemblages have been chosen from among all those shown in Tables 1 and 2. Regarding Plates 1 and 2 (by the way: why not to call them Figures?), they do not represent all recognized microfacies and micropaleontological assemblages: what are the adopted criteria? Why are they so relevant? Please explain. For example, you found calcareous turbidites and breccias, whose occurrence “in western Greece is still poorly constrained”, but they are poorly figured and described: why? As the paper largely regards Calcareous-Paleocene Units, this subject could be relevant. If it was the object of previous works, this should be stated.
- It is hard to really appreciate the results of this research. Nominally (“Results” paragraph), collected data allow to reconstruct vertical and lateral trends in facies architecture, with paleoenvironmental and evolutive implication for the Ionian Basin. In facts, the only result clearly presented here is the stratigraphic refinement, as well as the new proposed geological setting, of the study area, with implication for geological maps. All other results (tectono-sedimentary implication, paleogeographic restoration, etc.) are reported, but they are not enough supported by evidences. For example: how your data support the scheme of Figure 5? It is possible to propose analogous schemes divided by age? What about Late Jurassic, Cretaceous, etc. paleogeography of the area? What are the evidences that link Units to pre- and syn-rift phases? On what basis can be proposed a comparison with the other side of Adriatic and Ionian Seas (i.e.: the Apulian Platform and its margin)? And so on.
You can see the many perplexities raised by this work, which do not allow me to reach different conclusions. I'm sorry I can't evaluate this work differently at the moment; however, I hope my comments will be of some use to the authors.
Best Regards
The reviewer
